# Extreme Events Representation in CMCC-CM2 Standard and High Resolution General Circulation Models

Enrico Scoccimarro[1], Daniele Peano[1], Silvio Gualdi[1], Alessio Bellucci[2], Tomas Lovato[1], Pier Giuseppe Fogli[1] and Antonio Navarra[1]

[1]Fondazione Centro Euro-Mediterraneo sui Cambiamenti Climatici, Bologna, Italy
[2]Fondazione Centro Euro-Mediterraneo sui Cambiamenti Climatici, Bologna, Italy, currently at Consiglio Nazionale delle Ricerche, Istituto di Scienze dell'Atmosfera e del Clima, Bologna, Italy

*Correspondence to*: Enrico Scoccimarro (enrico.scoccimarro@cmcc.it)

**Abstract.** The recent advancements in climate modelling partially build on the improvement of horizontal resolution in different components of the simulating system. A higher resolution is expected to provide a better representation of the climate variability, and in this work we are particularly interested in the potential improvements in representing extreme events of high temperature and precipitation. The two versions of the CMCC-CM2 model used here adopt the highest horizontal resolutions available within the last family of the global coupled climate models developed at CMCC to participate in the CMIP6 effort.

The main aim of this study is to document the ability of the CMCC-CM2 models in representing the spatial distribution of extreme events of temperature and precipitation, under the historical period, comparing model results to observations (ERA5 Reanalysis, MSWEP and CHIRPS observations). For a more detailed evaluation we use both 6-hourly and daily time series, to compute indices representative of intense and extreme conditions.

In terms of mean climate, the two models are able to realistically reproduce the main patterns of temperature and precipitation.

The high resolution version (¼ degree horizontal resolution) of the atmospheric model provides better results than the standard resolution one (one degree), not only in terms of means but also in terms of intense and extreme events of temperature defined at daily and 6-hourly frequency. This is also the case of average and intense precipitation. On the other hand the extreme precipitation is not improved by the adoption of a higher horizontal resolution.

.

## 1 Introduction

An extreme climate event can have an impact on human activities, either as direct and indirect damages and, unfortunately also as loss of human life. For this reason it is very relevant to investigate General Circulation Models (GCMs) ability to simulate extreme events and to understand how the changing climate is influencing their distribution, frequency and location. Simulations of GCMs under the historical climate radiative forcing have been assessed in previous generations of the Coupled 30  Model Intercomparison Projects (CMIP; Flato et al. 2013) and, more recently, for CMIP6 (Eyring et al. 2016). Within CMIP6 the High Resolution Model Intercomparison Project protocol (HighResMIP, Haarsma et al., 2016) was designed to understand

the role of the horizontal resolution in improved process representation in all components of the climate system. In this paper, we present an analysis based on two versions of the GCM developed at CMCC (CMCC-CM, Cherchi et al., 2019), that we use for two simulations of the historical climate (1950-2014) differing only in their atmospheric horizontal resolution: HR with an horizontal resolution of 1 degree and VHR with a resolution of ¼ of a degree. The two models are described in detail in the next section.

The difference between the results obtained with the two versions of the model allows us to evaluate the impact of the model horizontal resolution on the temporal distribution of temperature and precipitation events compared to observations. It has been shown that the horizontal resolution can affect the representation of extreme events in state-of-the-art climate models (Van Haren et al., 2015; Iles et al., 2020). Besides, Demory et al. (2020) have shown that high-resolution models, when implemented with a resolution similar to VHR, achieve skills comparable to state-of-the-art Regional Climate Models in reproducing precipitation distributions over Europe. However, most of the analyses on extreme events employ rather low frequency data (typically daily), and short-duration high-intensity precipitation events can easily escape detection if high-frequency data are not used (Meredith et al. 2020, Scoccimarro et al. 2015).

Regarding the extremely high temperature representation, based on data at the daily frequency, it has been shown that GCMs tend to have warm bias over most land areas (Li et al., 2021) and the horizontal resolution plays a minor role in affecting the bias, with respect to the one played in the extreme precipitation representation (Kharin et al. 2013, Wei et al. 2019).

Regarding the extreme precipitation representation in GCMs, based on simulations from a single model, some improvement in skill at higher resolution for some measures of extreme precipitation over certain regions of the globe have been found in the past (Wehner et al. 2014, Kopparla et al. 2013). Only recently, multi-model assessment on this topic have been done, confirming that increasing the horizontal resolution to ¼ of degree (the highest adopted by the model object of this study), the magnitude of simulated daily (Bador et al. 2020) and sub-daily precipitation (Wehner et al. 2021) extremes is increased. On the other hand this is not associated to a systematic improvement in the simulation of precipitation extremes when compared to observations and, quantitatively, at the global scale, the intensification of precipitation extremes at increased resolution varies substantially from model to model (Bador et al. 2020). Also, for grid point GCMs (as opposed to spectral GCMs), the fraction of land precipitation increases, largely due to better resolved orography (Vannière et al., 2019; Terai et al., 2018; Demory et al., 2014).

In this paper we present both, a daily and a high-frequency analysis using 6-hourly data from the experiments, comparing model results to data from a reanalysis dataset with comparable horizontal resolution (ERA5, Hersbach et al. 2020) and two observational precipitation datasets such as MSWEP (Beck et al. 2019) and CHIRPS (Funk et al. 2015). The importance to evaluate extreme events at the sub-daily scale resides in the importance of such events on human health and over both urban and rural environments (Wehner et al. 2021).

The work is organized as follows: Sect. 2 describes the data and the methodology adopted, Sect.3 and Sect.4 describe the evaluation of model ability in representing the distribution of temperature and precipitation events respectively and Sect. 5 summarises and concludes the work.

## 2 Data and Methodology

### 2.1 The numerical experiments

The CMCC general circulation model has been developed in several configurations (Cherchi et al. 2019). The model uses as atmospheric module the CAM Atmospheric component (CAM4, Neale et al. 2010) in its grid point configuration. We will not go in a detailed description here, but since it is worthwhile to mention for our discussion on precipitation biases, the deep convection scheme is the one developed by Zhang and McFarlane (1995), modified following Ritcher and Rasch (2008) and Raymond and Blith (1986, 1992). The scheme is based on a plume ensemble approach where it is assumed that an ensemble of convective scale updrafts may exist whenever the atmosphere is conditionally unstable in the lower troposphere. Moist convection occurs only when there is convective available potential energy (CAPE) for which parcel ascent from the sub-cloud layer acts to destroy the CAPE at an exponential rate using a specified adjustment time scale. In other words the deep convection scheme is triggered based on a minimum positive threshold of CAPE, same as in the standard version of the CAM5 model (Wang and Zhang, 2013). The two models object of this study differ only in the horizontal resolution of their atmospheric component (CAM4) that is one degree in HR – the standard resolution one, and ¼ degree in VHR – the high resolution one. The ocean and sea-ice components are the same in HR and VHR models: a ¼ degree horizontal resolution version for both ocean (NEMO3.6, Madec & the NEMO team, 2016) and sea-ice (CICE4, Hunke & Lipscomb, 2008). The land model (CLM4.5, Oleson et al., 2013) is implemented with the atmospheric model grid. The basic of the coupling between the different components is described in Fogli and Iovino (2014). The single components of the coupled model are described in detail in Cherchi et al. (2019); additional studies based on last generation CMCC GCMs can be found in Scoccimarro et al. 2017a, Scoccimarro et al 2020, Bellucci et al. 2021. No changes are applied in terms of parameterization choices - and relative tuning parameters - moving from HR to VHR to be compliant with the HighResMIP protocol. Also, the two model versions use the same number of vertical levels in both atmosphere (26) and ocean (50) components. The complete set of experiments run with these two models is described in Haarsma et al. 2016. In the current analysis we investigate the hist-1950 HighResMIP experiment as described in section 2.3.

### 2.2 Re-analyses and observations for comparison

The model performance in representing the temperature distribution is evaluated by comparing results to the European Centre for Medium Range Weather Forecasts (ECMWF) ERA5 re-analyses (Hersbach et al. 2020, Andersson and Thepaut, 2008), with 137 hybrid sigma/pressure (model) levels in the vertical, and the top level at 0.01 hPa. The temperature data used in the paper (two-meter temperature, hereafter "temperature") can be obtained from the Copernicus Data Store (CDS) at https://cds.climate.copernicus.eu up to hourly frequency. The ERA5 horizontal resolution is about 0.28 degrees, close to the one of the higher (VHR) resolution model employed here (1/4 degree). It is important to note that the improvement of ERA5 reanalysis with respect to the previous ERA-Interim (Dee et al. 2011) product is due not only to the increased resolution but also to the addition of new integrated observation and aircraft data covering the recent decades, assimilated by the 4D-Var

algorithm. Since there are many known issues with ERA5 precipitation (Rivoire et al., 2021; Hu et al., 2020; Crosset et al. 2020), for the evaluation of the model performance in representing the precipitation distribution, we build on MSWEP version 2 observational dataset (Beck et al. 2019): The Multi-Source Weighted-Ensemble Precipitation (MSWEP) global precipitation dataset is available at a 3-hourly temporal resolution, covering the period from 1979 to the near present, with an horizontal resolution of 0.1 degrees. The dataset takes advantage of the complementary strengths of gauge-, satellite-, and reanalysis-based data to provide reliable precipitation estimates over the globe.

Since we aim to characterize different types of extreme events, we consider both 6-hourly and daily time series for the computation of the percentiles (see 2.3) for the chosen climate parameters.

For a more exhaustive evaluation of the precipitation distribution, we also take advantage of the CHIRPS (Climate Hazards group Infrared Precipitation with Stations) daily observational dataset. The version 2.0 of the CHIRPS database comprises a quasi-global (50°S-50°N, 180°E-180°W) domain, at ¼ degree resolution, and 1981 to near-present gridded precipitation daily time series. This dataset merges three types of information: global climatology, satellite estimates, and in situ observations (Funk et al. 2015).

## 2.3 Methodology

The period used to compare the simulated temperature (tas) distribution to the observations is 1950-2014. On the other hand, due to the shorter period available for the MSWEP and CHIRPS datasets, the precipitation (pr) distribution is evaluated over the common period between the observations and the historical model run 1981-2014. This time period is sufficiently long to capture the temporal variability at the global scale (Schindler et al. 2015). Typically, the warm extremes are computed based on maximum daily temperature, but in this work we want to verify the potential improvements induced by the increased resolution in the representation of extreme temperature events defined at two different time frequency (daily and 6-hourly). For this reason we investigate the distribution of daily and 6-houry average temperature (tas), instead of maximum daily temperature.

Model averages and 99[th]/90[th] percentile (99p/90p hereafter) are computed on the native grid and then the results are compared to ERA5 or observational datasets, linearly interpolating the re-analysis (or observations) on the model grid. The kind of interpolation introduces very little differences in the fields (not shown). We denote events belonging to the 99p as "extreme events" and the ones belonging to the 90p as "intense events" (Scoccimarro et al. 2016). Two seasons are considered, December to February (DJF hereafter) and June to August (JJA hereafter) representative of the boreal winter and summer, respectively.

Temperature percentiles computed at the daily time frequency are obtained based on a sample of 5850 (*90 days x 65 years)* events, while the percentiles computed at the six-hourly time frequency are obtained based on a sample of 23400 (*90 days x 65 years x 4 six-hourly data in a day*) events. Precipitation percentiles computed at the daily time frequency are obtained based on a sample of 3060 (*90 days x 34 years)* events, while the percentiles computed at the six-hourly time frequency are obtained based on a sample of 12240 (*90 days x 34 years x 4 six-hourly data in a day*) events.

Temperature related parameters are expressed in degree Celsius [°C], and precipitation related parameters in [mm/d] . When expressed as % fraction (Figure S17 only) the precipitation is shown only for regions where the seasonal average of precipitation is higher than 0.5 mm/d to avoid misleading percentual differences over dry domains (Scoccimarro et al. 2013). The comparison with CHIRPS precipitation data is performed at the daily frequency only.

## 3 Representation of extreme events of temperature

In this section modelled extreme temperature is compared to the ERA5 reanalysis. Figure 1 shows the DJF 99th percentile of ERA5 temperature time series (upper panel), at the daily frequency, together with model results (central panels) and relative biases (lower panels). Figure 2 shows the JJA season results following the same structure while Figure 3 and Figure 4 refer to 6-hourly statistics. Higher values for extreme events appear when focusing on the 6-hourly results, with maximum differences
(up to 5 °C) along the Tropics and in particular over central America, western India and equatorial Africa during DJF (Figure 1 compared to Figure 3, upper panels) and over northern Africa, Saudi Arabia and western United States during JJA (Figure 2 compared to Figure 4, upper panels).

The daily based extreme temperature bias is shown in figure 1 (for DJF) and figure 2 (for JJA) for the HR and VHR models in the lower panels. The large positive DJF bias shown by the HR model at the high latitudes in the Northern Hemisphere -
145 reaching 9 °C over Alaska, northern Canada and eastern Siberia (Figure 1 lower left panel) - is significantly reduced in the VHR model (Figure 1, lower right panel). Also the positive HR DJF bias over eastern Europe is more than halved in VHR, while the DJF negative biases over northern Africa and Tibetan Plateau worsen moving to the higher resolution. The positive extreme temperature bias between 30°N and 60°N shown by the HR model during JJA (Figure 2 lower left panel) is partially reduced in VHR especially over Europe and Asia. Similarly, the 5 to 7°C positive JJA bias over the western coast of South
America in HR, results halved in VHR. On the other hand the negative JJA bias of about -8°C over north-eastern Canada shown by HR model is even worse in the VHR version, where a larger portion of the domain is subject to a bias of about -10°C. This negative bias is also consistent with the tendency of the two versions of the CMCC-CM2 model to overestimate the sea ice cover during summer over the Northern Hemisphere (not shown).

Moving to the 6-hourly based extreme events, the fraction of land affected by a positive bias higher than 5°C is more
pronounced compared to the daily statistics, especially for the HR model during JJA (Figure 4). The positive bias over the north western part of South America, during JJA, reaches 9°C in HR and is only partially reduced in VHR; during the same season the positive bias of the same order of magnitude over central and eastern United States is not improved by the increased resolution. Similar patterns, but less pronounced, are reflected on the averaged temperature, as shown in supplemental figures S1-S2, and  intense events representation (Figures S7-S10).

## 4 Representation of extreme events of precipitation

Following the same structure as in the previous section, the model extreme precipitation is here compared to the MSWEP observations (from Figure 5 to Figure 8) for both daily and 6-hourly statistics, and then to the CHIRPS dataset (figures 9 and 10) for daily statistics only. Figure 5 shows the MSWEP DJF seasonal extreme precipitation (upper panel) during the historical period and the modeled results (central panels) together with the relative biases (lower panels). Figure 6 shows the same 99p parameter but for JJA, computed based on daily time series. Figures 7 (for DJF) and 8 (for JJA), instead, show the 99p computed based on 6-hourly time series. The higher extreme events magnitude associated to the 6-hourly results (Figures 7 and 8, upper panel) compared to the daily statistics (Figures 5 and 6 , upper panel) is visible almost everywhere, but it is more pronounced over the Tropics. In fact this is where convective processes are expected, and it is well known that convective precipitation tends to be short lived, while long-duration intense events (from 12 hours to 3 day) are often associated to synoptic weather systems and tend to have larger spatial scales (Chan et al. 2014, Scoccimarro et al. 2015).

In terms of average precipitation the VHR model shows less pronounced biases with respect to HR model (Figures S3 and S4 for DJF and JJA respectively based on MSWEP and Figures S5 and S6 for the same seasons based on CHIRPS). In particular, during DJF, the negative bias over northern part of South America is reduced from about 4 to 2 mm/d, while the positive bias over western United States, South Africa and Australia is almost halved. During JJA, the bias tends to be less pronounced in both models, and the differences between the two are mainly located over Peru, Bolivia and Brazil ranging from about -3 mm/d of the HR model to values closer to zero, even positive, over a small portion of the domain in the VHR model.

A different behavior is found focusing on daily extreme precipitation events. No particular differences between high and low resolution biases are found north of 30ºN during winter (Figure 5), while the VHR model tends to overestimate the 99[th] percentile of daily precipitation distribution in both seasons within the Tropics (Figures 5 and 6). Similar patterns emerge for the 6-hourly based extreme precipitation (Figure 6), but with a less pronounced overestimate in VHR over the Tropics, compared to HR results. The intense events are better represented by the VHR model compared to the HR one, especially during winter in the Southern Hemisphere (Figure S11, lower panels), where the 8 mm/d HR positive bias over Australia and South Africa is halved in VHR. This is consistent with the better representation of the DJF average precipitation in the VHR model (Figure S3), suggesting that the bad representation of DJF extreme precipitation in VHR (Figure 5) is mainly due to a too much pronounced stretching of the right part of the precipitation distribution.

To corroborate our results in terms of precipitation biases, we computed the same statistics obtained from MSWEP, using the CHIRPS observational daily dataset for averages (Figure S5 and S6) and extreme events (Figures 9 and 10). The biases computed with respect to the CHIRPS dataset are very similar to what we already described based on MSWEP, but with a slightly increased magnitude (Figure 9 compared to Figure 5) for extreme events in both models, especially during DJF, along the Tropics.

The worsening of the extreme precipitation bias moving from the HR to the VHR model along the tropics, especially in the Southern Hemisphere during JJA, is also associated to a deterioration of the representation of the fraction of precipitation

associated to extreme events with respect to the total precipitation: this metric is obtained accumulating the water of all the events more intense than the 99p, and normalizing it by the total amount of precipitation in the considered period (season by season). Figure S17 shows that both models reasonably well capture this metric in both seasons compared to MSWEP, but the VHR model tends to overestimate such amount over the Southern Hemisphere, except for the Australian domain. In particular, the strong positive bias of DJF average precipitation over Australia (up to 4 mm/d, Figure S3, lower panels) can't be attributed to the positive (higher than 15 mm/d, Figure 5 lower panels) bias found for extreme events, but must be associated to a right shift of the remaining part of the precipitation distribution, more pronounced for the non-extreme events as partially confirmed by the positive bias in the 90p metric over the same season (Figure S11).

## 5 Summary and conclusions

CMCC-CM2-HR4 and CMCC-CM2-VHR4 models are state-of-the-art fully coupled climate models, participating in different Model Intercomparison Projects within the 6th Coupled Model Intercomparison Project (CMIP6). CMCC-CM2-HR4 presents a horizontal resolution typical of most of the CMIP6 involved models, while CMCC-CM2-VHR4 has a horizontal resolution standard for the models involved in the High-Resolution Model Intercomparison Project (HighResMIP). In this paper we highlight the ability of the two models to represent extreme climate conditions, based on daily and 6-hourly time series, comparing temperature and precipitation modelled distributions to the observed ones. In order to have a gridded dataset representative of the observed climate at the daily and 6-hourly time frequency we used ERA5 reanalysis for temperature and MSWEP observations for precipitation. For the precipitation analysis we also reinforce our investigation on the base of the CHIRPS daily observations.

It is well known that the representation of precipitation extreme indices is more dependent on the horizontal resolution than what we expect for temperature extreme indices (Wei et al. 2019). Anyway, on average, the highest resolution CMCC model (VHR) is better than the lower resolution model (HR) in representing average, intense (90p) and extreme (99p) events of temperature both in terms of patterns and magnitude. This is true for daily and 6-hourly based statistics. Also VHR results are quite in agreement with CMIP6 multi-member average of daily intense and extreme temperature indices (Scoccimarro and Navarra, 2021). The described differences between the computed daily and 6-hourly biases in temperature statistics are very similar for HR and VHR models. This result suggests that a higher horizontal resolution is not sufficient to improve the representation of extreme temperature events at the highest time frequency considered. Consequently, the worsening of model biases in high frequency (6-hourly) temperature statistics derives from deficiencies of the current version of model components and parameterizations in representing high-frequency processes.

Regarding the precipitation distribution, the VHR model performs better in representing averages and intense events, but more pronounced biases appear in VHR compared to HR when focusing on extreme events, with a more evident degradation in the daily statistics compared to the 6-hourly. This latter result reduces the confidence we usually attribute to the highest horizontal

resolution in modelling extreme precipitation, and is consistent with single model analysis based on CAM5.1 atmospheric model (Wehner et al. 2014) suggesting a positive bias over most of the globe in the representation of extreme events at ¼ degree horizontal resolution. This is also in agreement with recent findings (Bador et al. 2020) suggesting that highest resolution models tend to produce more pronounced extremes than lower resolution ones. In addition many of them show lower skill in representing observed patterns, both in terms of intensity and spatial distribution, at the higher resolution,

compared to their corresponding lower resolution version.

This emphasizes the need to focus not only on the horizontal resolution to improve the model ability in representing the climate system, but also on physics and tuning. It is important to note that in the model object of this analysis the tuning parameters were kept constant, moving from the HR to the VHR version, in order to be compliant with the HigResMIP protocol.

The different biases, obtained based on daily and 6-hourly time frequencies, also suggest that for the setup of model physics

and tuning we need to consider the event distributions at different time frequencies, to take into account the representation of the different processes responsible of the extreme conditions emerging at the different frequencies (Scoccimarro et al. 2015).

The poor performance of climate models in representing extreme precipitation is not improved in the last CMIP6 generation models, compared to the previous CMIP5 generation (Scoccimarro et al. 2020). In the present work we have shown that this lack is even more evident moving to the highest resolution version of the CMCC-CM2 model adopted for HighResMIP,

consistently with multi-model analysis performed at the same horizontal resolution (Bador et al. 2020): GCMs whose parameterizations were not retuned at higher resolution lead to worse results. The high-resolution version of the model  tends to overestimate extreme precipitation in the wet and warm regions, consistently with findings based on experiments carried out with the CAM5 atmospheric model at the same resolutions (Wehner et al, 2014), highlighting once again  the importance of an extensive model tuning at the high resolution. In addition it is important to note that moving from the standard to the

high resolution of CMCC-CM2, the model behaves consistently with the models participating to the HighResMIP project: a tendency to an increased fraction of land precipitation in the highest resolution, and the same tendency for the fraction of land precipitation caused by moisture convergence (Venniere et al. 2019). Also, in CMCC-CM2 model,  the orographic precipitation captures most of the change of precipitation due to resolution, consistently with most of HighResMIP models (Venniere et al. 2019).

In principle, the horizontal resolution increase should improve the representation of extreme storms such as tropical cyclones (Scoccimarro et al. 2020) and for this reason also the representation of the associated short term extreme precipitation should improve, but this is not the case for the model object of this study, and it is also confirmed by recent analysis on the same topic (Wehner et al., 2021).


**Code and Data availability**

The code relative to the CMCC-CM2-HR4 and the CMCC-CM2-VHR4 climate models is available on the Zenodo repository (URL: https://zenodo.org/record/5499856#.YTs5Bh2xVZP, doi: 10.5281/zenodo.5499856). The data relative to the two models are available through the ESGF data portal (Scoccimarro et al. 2017b and Scoccimarro et al. 2017c,

respectively). ERA5 Reanalysis are available through the Copernicus data portal (https://climate.copernicus.eu). CHIRPS observational dataset is available through the data storage of the University of California in Santa Barbara (https://www.chc.ucsb.edu/data/chirps).

**Author contribution**

ES, AB and DP implemented the two model versions and run the simulations. PGF supported the implementation of the Aerosol input management routines.TL prepared the radiative forcing files and supported the model output postprocessing. ES prepared the manuscript with contributions from all co-authors.

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

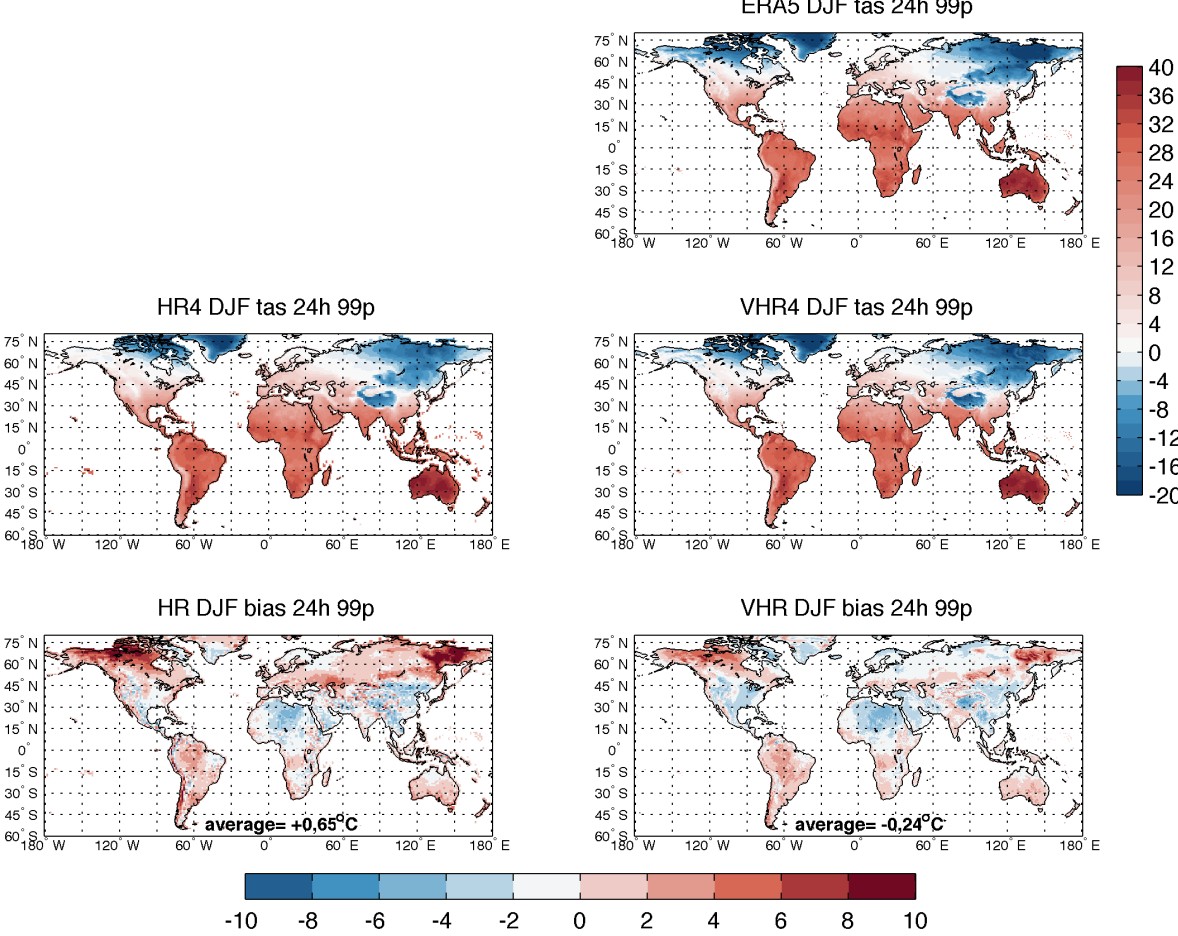

**Figure 1: Winter (DJF) Extreme Temperature (99th percentile, 99p) computed at the daily frequency. Upper panel shows ERA5 results. Central panels show model results (HR on the left and VHR on the right) and lower panels show the relative model bias. Units are [ºC]. Vertical colorbar refers to the three upper panels, while the horizontal colorbar refers to the two bottom panels.**

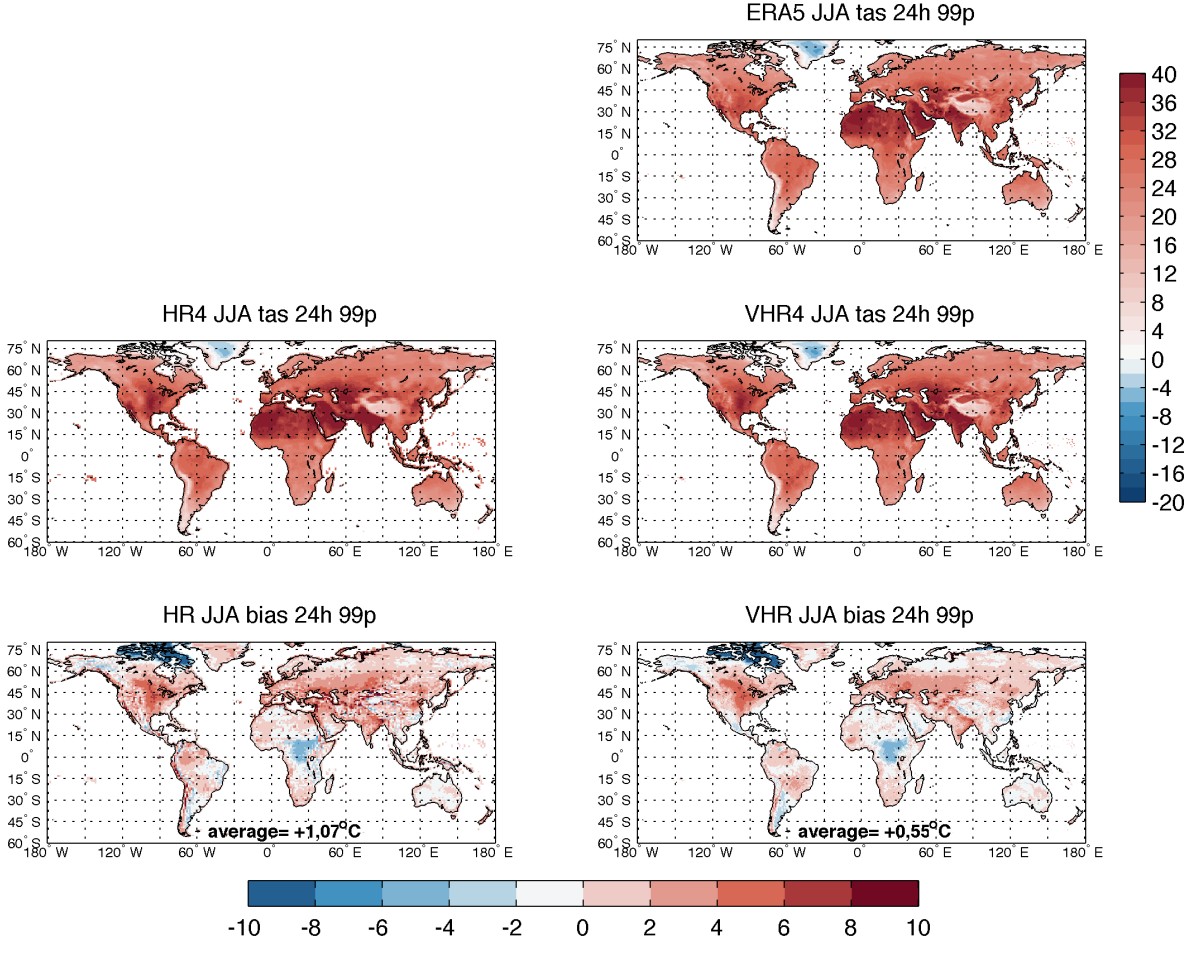

**Figure 2: Summer (JJA) Extreme Temperature (99th percentile, 99p) computed at the daily frequency. Upper panel shows ERA5 results. Central panels show model results (HR on the left and VHR on the right) and lower panels show the relative model bias. Units are [ºC]. Vertical colorbar refers to the three upper panels, while the horizontal colorbar refers to the two bottom panels.**


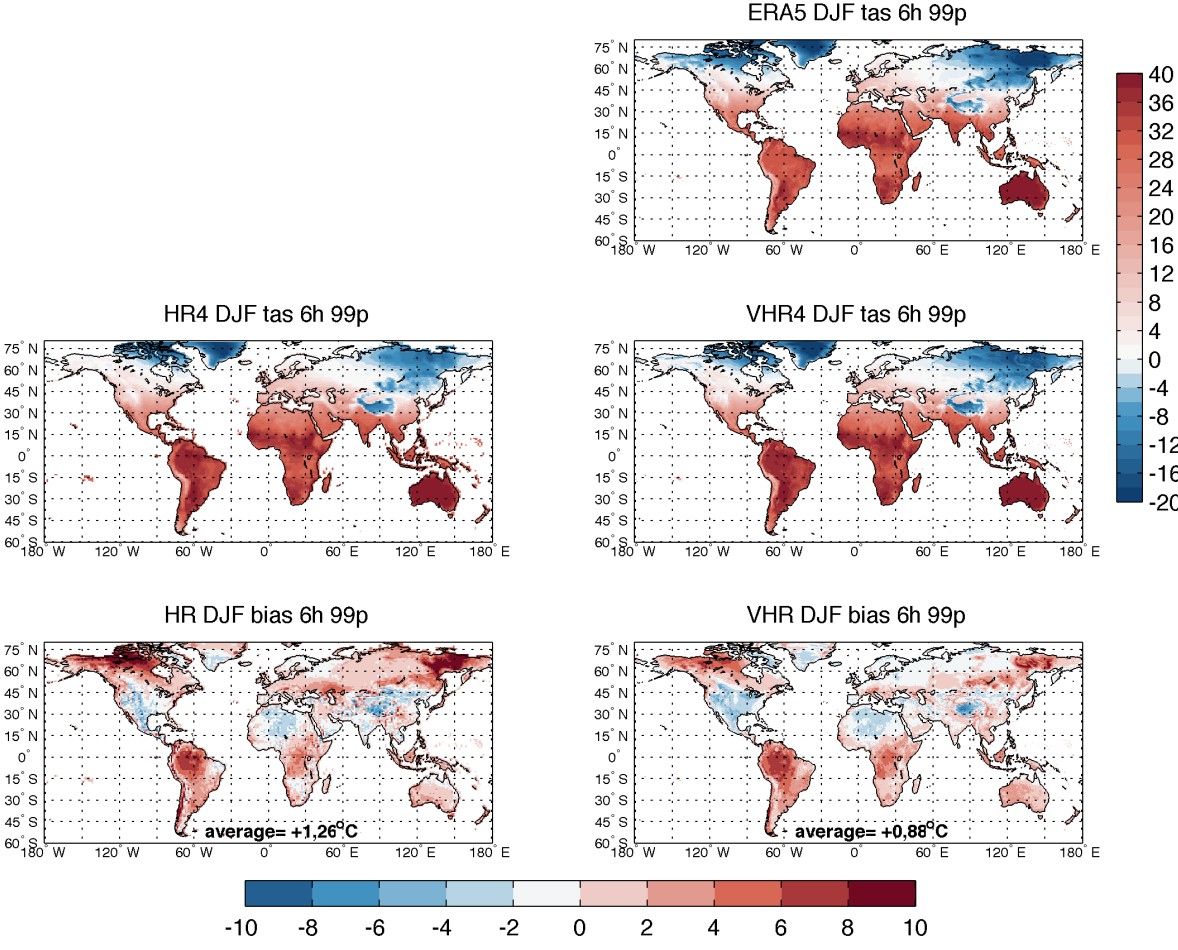

**Figure 3: Winter (DJF) Extreme Temperature (99th percentile, 99p) computed at the six-hourly frequency. Upper panel shows ERA5 results. Central panels show model results (HR on the left and VHR on the right) and lower panels show the relative model bias. Units are [ºC]. Vertical colorbar refers to the three upper panels, while the horizontal colorbar refers to the two bottom panels.**

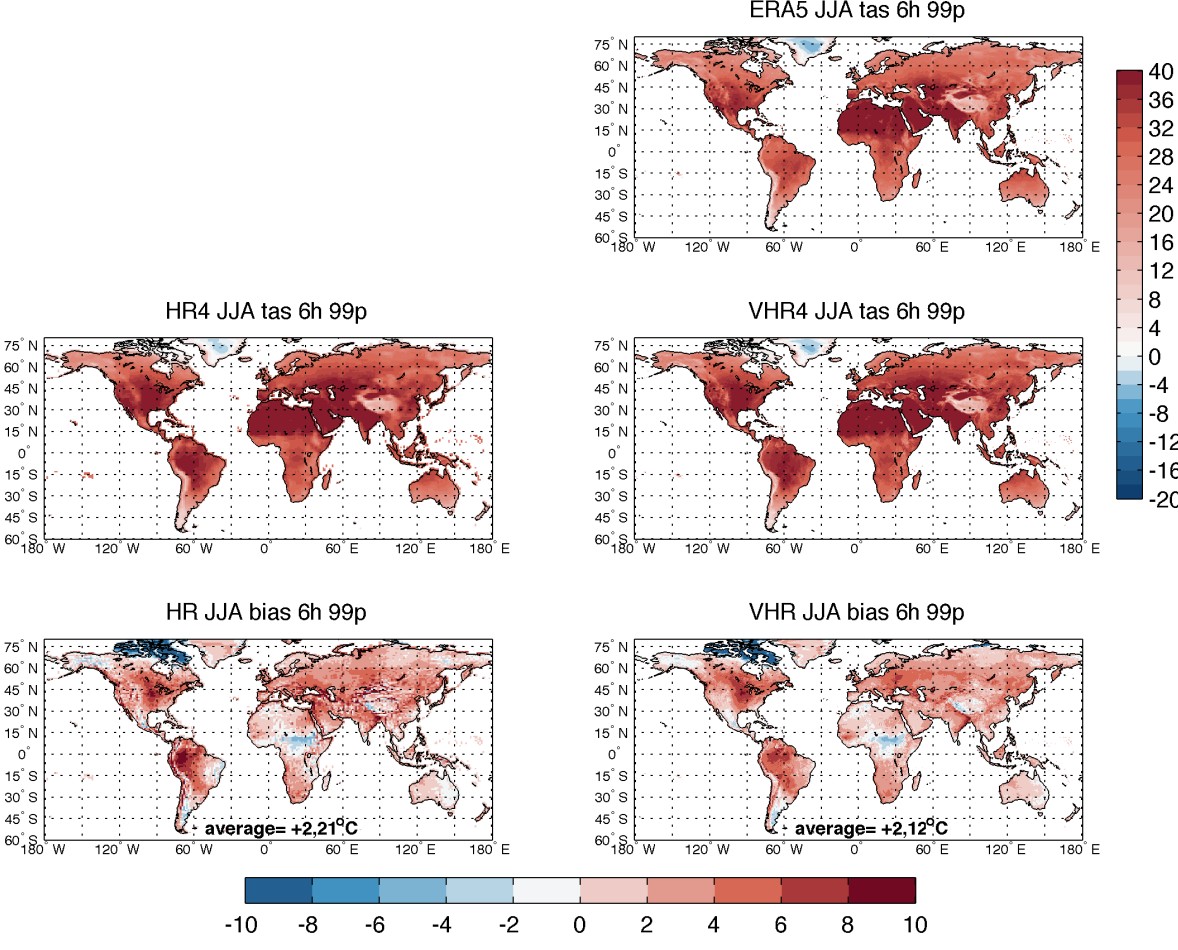

Figure 4: Summer (JJA) Extreme Temperature (99th percentile, 99p) computed at the six-hourly frequency. Upper panel shows ERA5 results. Central panels show model results (HR on the left and VHR on the right) and lower panels show the relative model bias. Units are [ºC]. Vertical colorbar refers to the three upper panels, while the horizontal colorbar refers to the two bottom panels.


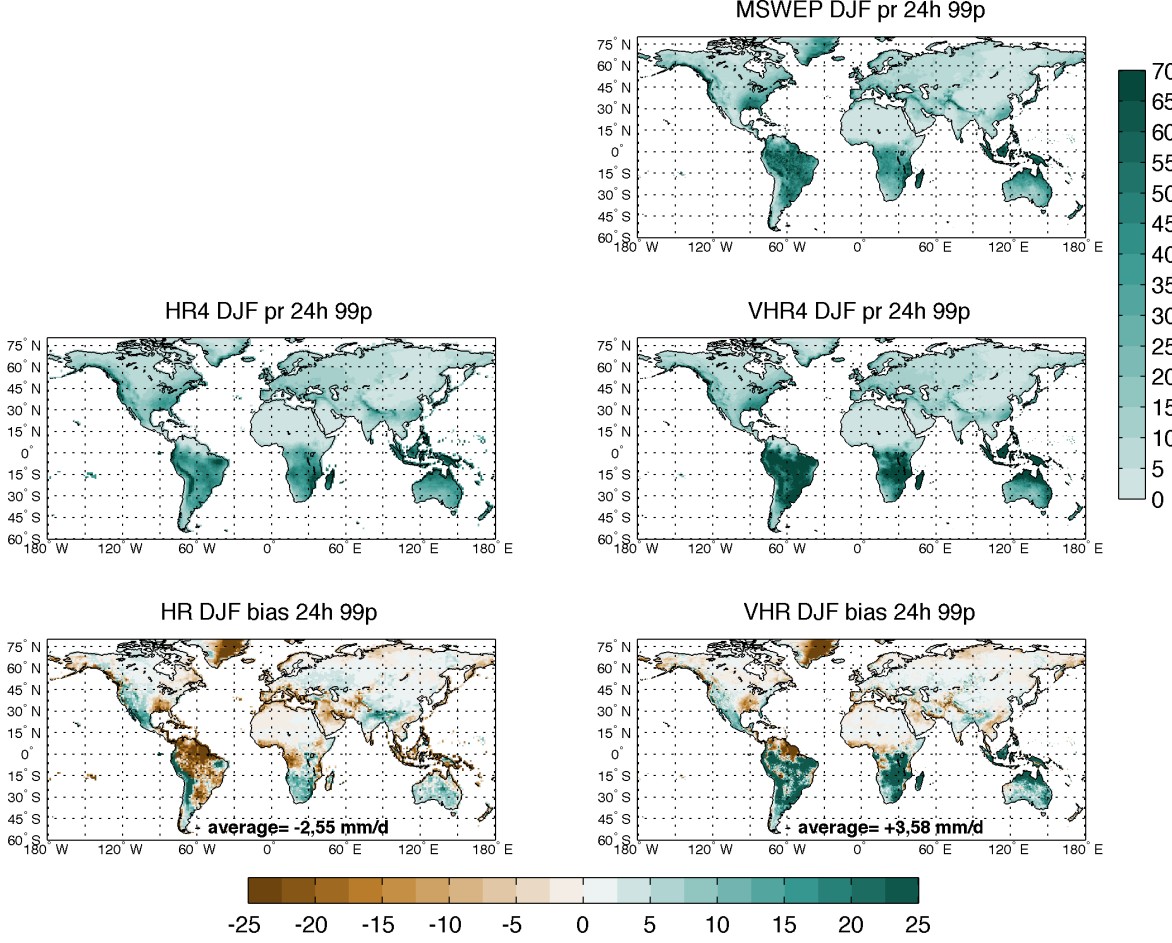

**Figure 5: Winter (DJF) Extreme Precipitation (99th percentile, 99p) computed at the daily frequency. Upper panel shows MSWEP observational results. Central panels show model results (HR on the left and VHR on the right) and lower panels show the relative model bias. Units are [mm/d]. Vertical colorbar refers to the three upper panels, while the horizontal colorbar refers to the two bottom panels.**

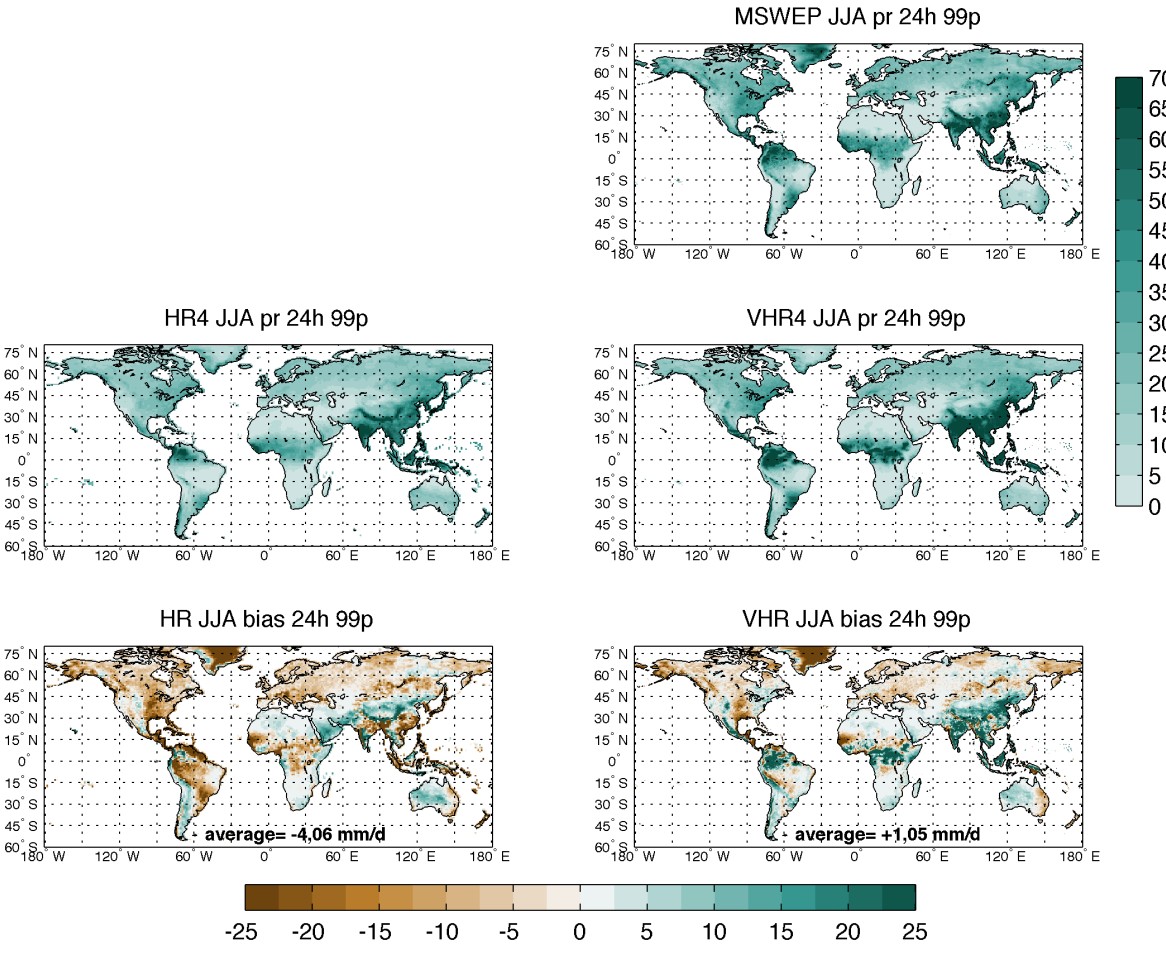

**Figure 6: Summer (JJA) Extreme Precipitation (99th percentile, 99p) computed at the daily frequency. Upper panel shows MSWEP observational results. Central panels show model results (HR on the left and VHR on the right) and lower panels show the relative model bias. Units are [mm/d]. Vertical colorbar refers to the three upper panels, while the horizontal colorbar refers to the two bottom panels.**

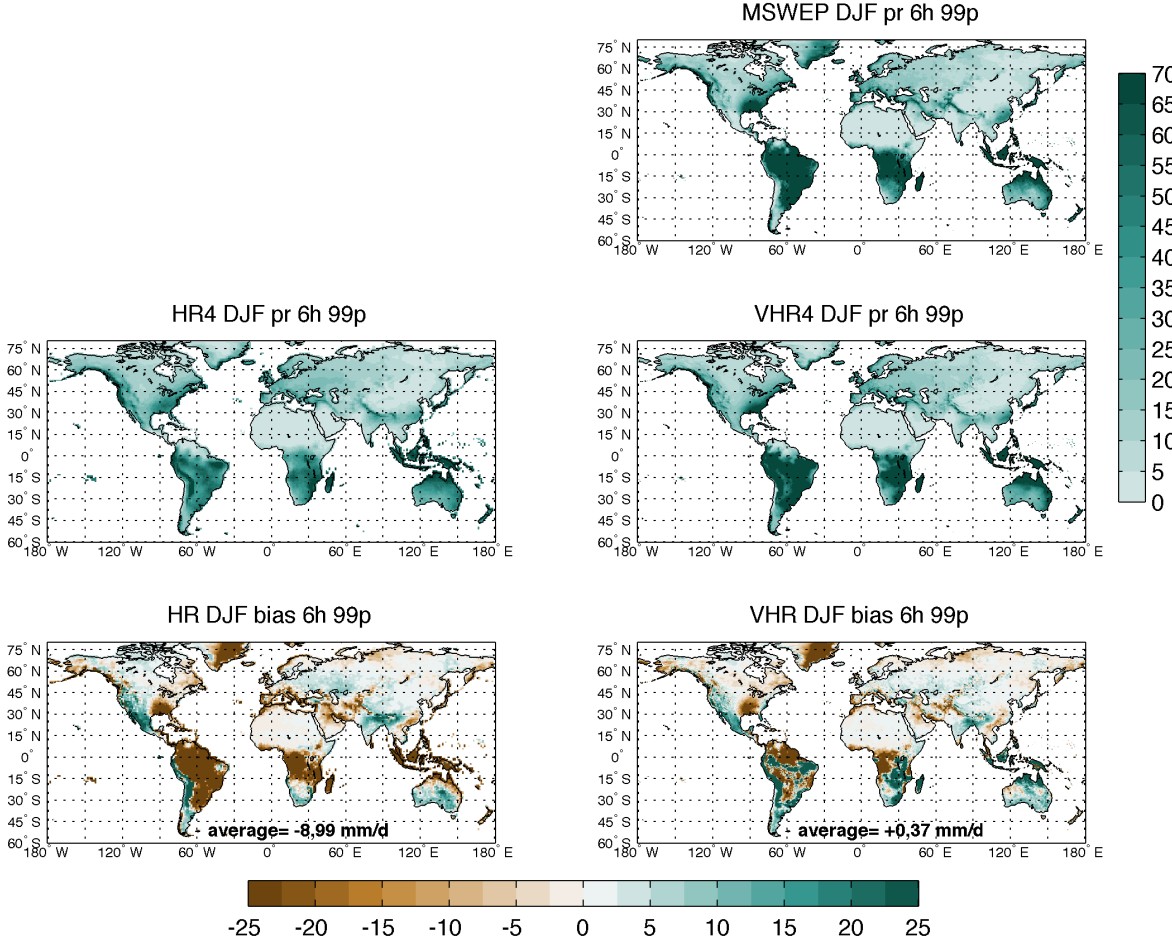

**Figure 7: Winter (DJF) Extreme Precipitation (99th percentile, 99p) computed at the six-hourly frequency. Upper panel shows MSWEP observational results. Central panels show model results (HR on the left and VHR on the right) and lower panels show the relative model bias. Units are [mm/d]. Vertical colorbar refers to the three upper panels, while the horizontal colorbar refers to the two bottom panels.**

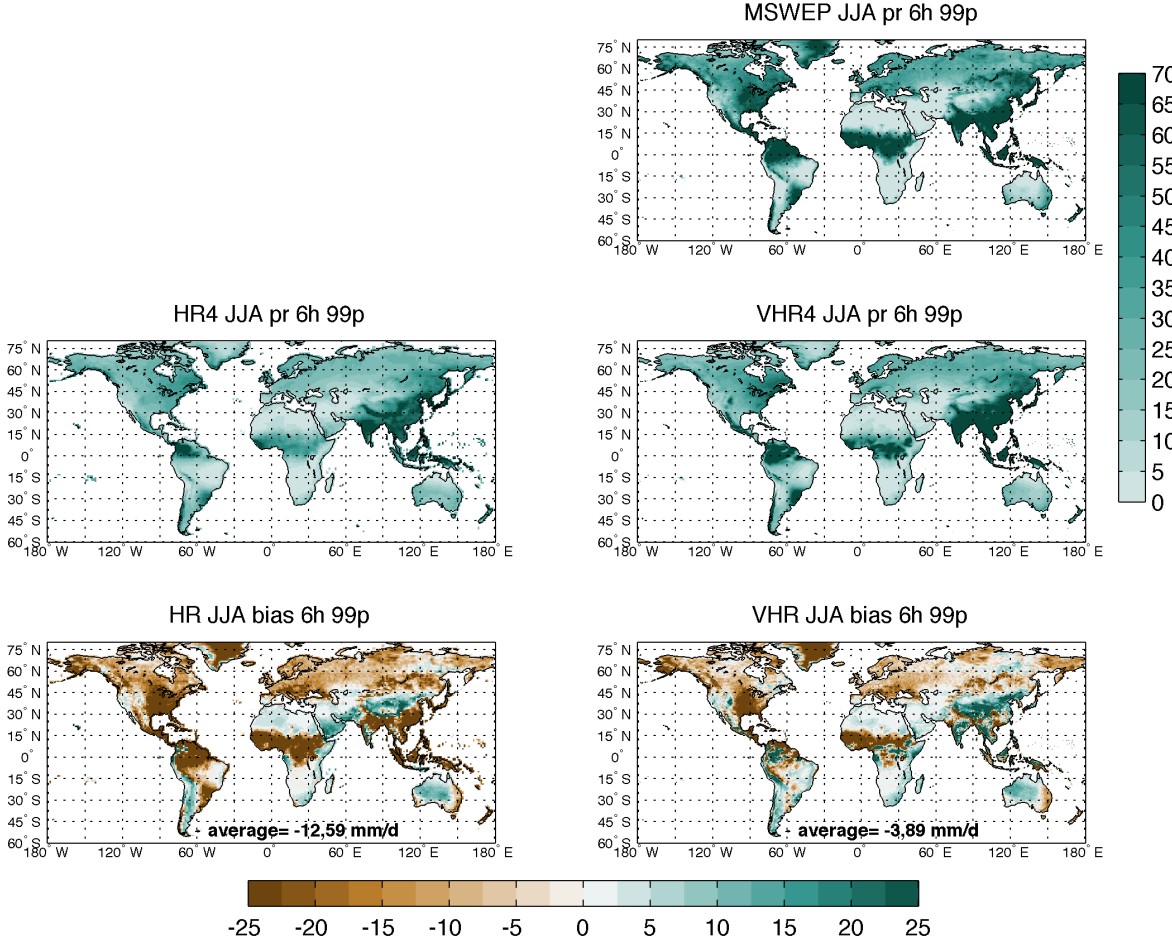

**Figure 8: Summer (JJA) Extreme Precipitation (99th percentile, 99p) computed at the six-hourly frequency. Upper panel shows MSWEP observational results. Central panels show model results (HR on the left and VHR on the right) and lower panels show the relative model bias. Units are [mm/d]. Vertical colorbar refers to the three upper panels, while the horizontal colorbar refers to the two bottom panels.**

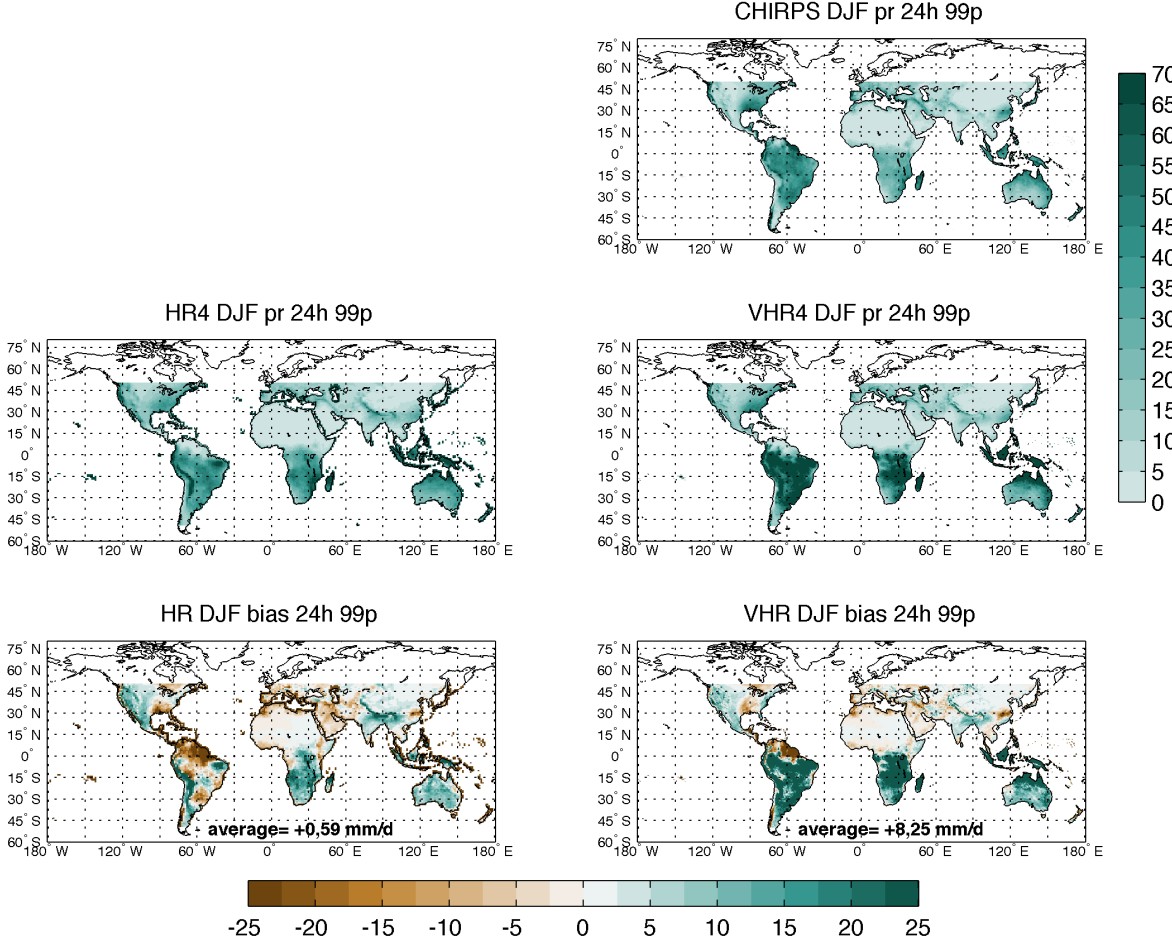

Figure 9: Same as figure 5 but based on CHIRPS observations. Winter (DJF) Extreme Precipitation (99th percentile, 99p) computed at the daily frequency. Upper panel shows CHIRPS observational results. Central panels show model results (HR on the left and VHR on the right) and lower panels show the relative model bias. Units are [mm/d]. Vertical colorbar refers to the three upper panels, while the horizontal colorbar refers to the two bottom panels.

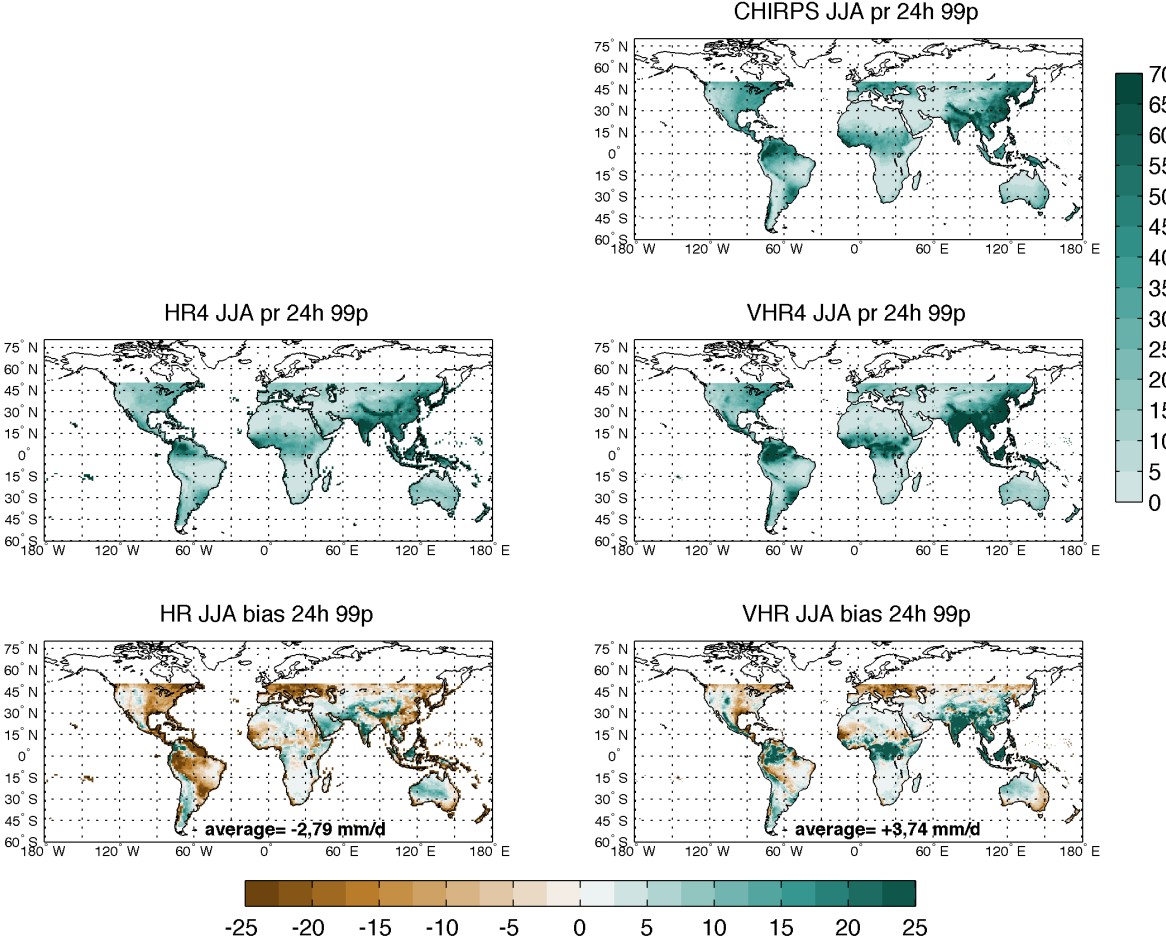

**Figure 10: Same as figure 6 but based on CHIRPS observation. Summer (JJA) Extreme Precipitation (99th percentile, 99p) computed at the daily frequency. Upper panel shows CHIRPS observational results. Central panels show model results (HR on the left and VHR on the right) and lower panels show the relative model bias. Units are [mm/d]. Vertical colorbar refers to the three upper panels, while the horizontal colorbar refers to the two bottom panels.**

485

490