# Peer review of "Extreme Events Representation in CMCC-CM2 Standard and High Resolution General Circulation Models"

_Geoscientific Model Development, 2021_

## Author Comment (AC1)

**Answers to RC1:**
We thank the reviewer for the suggestions and we answered (in blue) to each single comment (in black), focusing on the best solution to satisfy RC1 and RC2 comments (both available on the online discussion).

The paper evaluates the impact of an increase of horizontal resolution on the representation of extreme events in CMIP5 and HighResMIP type models. I think that this could become a valid contribution, but I would like to encourage the authors to extend the evaluation and to provide additional information and plots as outlined in the comments below.

As already stated in the introduction we tried to match as much as possible RC1 suggestions in the new version of the manuscript.

Major:

- The use of "high" resolution for 1 degree and "very high" resolution for ¼ degree is misleading. Why not use "standard" and "high" resolution. Otherwise, you would need to come up with "extremely" high and "ultra" high for the next resolution upgrades.

  In the new version of the manuscript we use the terms "standard" and "high" instead of "high" and "very-high".

- To my best knowledge, ERA5 does not assimilate precipitation and precipitation is only available as a diagnostic of 18h forecasts that are used for data assimilation. This is not a big problem, but could you please specify in the text where the precipitation field is coming from? Also, the grid and resolution of ERA5 should be specified (I think it is not ¼ degree).

  In the new version of the manuscript we do not rely on ERA5 precipitation for model evaluation. Also following the Reviewer #2 comment, we decided to use a new observational dataset (in addition to the already involved CHIRPS dataset) instead of ERA5. The MSWEP (Beck et al. 2019) dataset is a global precipitation product with a 3-hourly 0.1° resolution available at a 3-hourly temporal resolution, covering the period from 1979 to the near present. The dataset takes advantage of the complementary strengths of gauge-, satellite-, and reanalysis-based data to provide reliable precipitation estimates over the globe. With this dataset we compute seasonal averages and both daily and 6-hourly percentiles to evaluate model results (same as done based on ERA5 in the previous version of the paper, but over a shorter period [1981-2014).

  The ERA5 resolution is now made explicit in section 2.2.

  *References:*
  *Beck, H. E., Wood, E. F., Pan, M., Fisher, C. K., Miralles, D. G., van Dijk, A. I. J. M., McVicar, T. R., & Adler, R. F.. MSWEP V2 Global 3-Hourly 0.1° Precipitation: Methodology and Quantitative Assessment, Bulletin of the American Meteorological Society, 100(3), 473-500, 2019.*

- I would recommend comparing against two observation datasets for precipitation. This would allow to judge how much (or little) we actually know about mean and extreme precipitation fields.

  As introduced in the previous comment we added a new precipitation dataset (MSWEP, Beck et al. 2019) for the evaluation of the model ability in representing averages, extreme and intense events at the 6-hourly and daily time frequency.

- The evaluation of extreme events is interesting. But please also discuss the mean temperature and precipitation fields in more detail and include plots of the mean fields and biases. It is difficult to judge the quality of the representation of extreme events if the quality of the mean field representation is unclear. In particular, as you are referring to average representations for temperature and precipitation in the summary.

  Temperature and precipitation averages, commented in the manuscript, are shown in supplemental figures S1-S6, now defined following the reviewer suggestion to "put the reference and the bias fields into the same figure" keeping separate "6h/24h and DJF/JJA" fields (see the last RC1 major comment). Be aware that, following the reviewer #2 suggestion, we had to change the color schemes used, to present the data based on the IPPC visual style guide.

- Why are you focussing so much on the 99th Please also add plots and discussion of the 90th percentile.

  We added maps of the 90th percentiles for temperature (figures S7-S10) and precipitation, computed with MSWEP (daily S11-S12 and 6-hourly S13-S14) and CHIRPS (only at the daily frequency, S15-S16). These figures are now considered when commenting intense events in sections 3 and 4.
  Old figure S4 is now S17.

- There should be more discussion on the impact of a resolution upgrade on temperature and precipitation predictions that was observed by other modelling groups.

  Regarding the dependence of the extreme temperature representation on the horizontal resolution, the relative summary section has been modified as:
  "*It is well known that the representation of precipitation extreme indices is more dependent on the horizontal resolution than what we expect for temperature extreme indices (Wei et al. 2019). Anyway, on average, the highest resolution CMCC model (VHR) is better than the lower resolution model (HR) in representing average intense (90p) and extreme (99p) events of temperature both in terms of patterns and magnitude. This is true for daily and 6-hourly based statistics. Also VHR results are quite in agreement with CMIP6 multi-member average of daily intense and extreme temperature indices (Scoccimarro and Navarra, 2021)*"

  Regarding the dependence of the extreme precipitation representation on the horizontal resolution, the relative summary section has been modified as:
  "*Regarding the precipitation distribution, the VHR model performs better in representing averages and intense events, but more pronounced biases appear in VHR compared to HR when focusing on extreme events, with a more evident degradation*

*in the daily statistics compared to the 6-hourly. This latter result reduces the confidence we usually attribute to the highest horizontal resolution in modelling extreme precipitation, and is consistent with single model analysis based on CAM5.1 atmospheric model (Wehner et al. 2014) suggesting a positive bias over most of the globe in the representation of extreme events at ¼ degree horizontal resolution, and also with multi-model recent findings (Bador et al. 2020) suggesting that highest resolution models tend to produce more pronounced extremes than lower resolution ones. . In addition many of them show lower skill in representing observed patterns, both in terms of intensity and spatial distribution, at higher resolution compared to their corresponding lower resolution version."*

Added references:
Scoccimarro E., Navarra A.: Precipitation and temperature extremes in a changing climate. Chapter 2 in "Hydrometeorological Extreme Events and Public Health" Wiley book, 320 pages. ISBN: 978-1-119-25930-5.

Wehner MF et al.: The effect of horizontal resolution on simulation quality in the Community Atmospheric Model, CAM5.1. J. Adv. Model. Earth Syst. 6, 980–997. doi:10.1002/2013MS000276, 2014.

Wei, L. X., X. G. Xin, C. Xiao, et al.: Performance of BCC-CSM Models with Different Horizontal Resolutions in Simulating Extreme Climate Events in China. *J. Meteor. Res.*, 33(4): 720-733. doi: 10.1007/s13351-019-8159-1, 2019.

There should also be a discussion how the move to storm-resolving resolutions could change the situation.
We added the following sentence to the summary section:
"In principle, horizontal resolution increases should improve the representation of extreme storms such as tropical cyclones (Scoccimarro et al. 2020) and for this reason also the representation of the associated short term extreme precipitation should improve, but this is not the case for the model object of this study, and it is also confirmed by recent studies on the same topic (Wehner et al. 2021)."

Furthermore, I would like to know how the parametrisation schemes change when resolution is increased from 1 degree to ¼ degree.
No change was applied, to follow the PRIMAVERA (EU project) protocol, as now specified in section 2.1 on *numerical experiment description*.

- It is very hard to compare the fields in the figures at the moment. Please put the model fields, the reference (ERA or observations) and the bias fields into the same figure. You can separate 6h/24h and DJF/JJA. At the moment, a lot of flipping through the pages is required to compare the fields.
  This was our first choice when starting to collect and present our results, but then we moved to the "shorten" version you have seen, for readability. Anyway, to satisfy the reviewer request, in the new version of the manuscript we went back and all the figures are presented following the suggestion to put the model fields, the reference and the bias fields into the same figure, separating 6h/24h and DJF/JJA. Adding also

the 90th percentile maps we now have 10 figures in the main manuscript and 17 supplemental figures, where S7-S16 refer to the 90th percentile.

Minor:

l17: "for the definition of the extreme condition" Please re-word

Rewritten as: "For a more detailed evaluation we use both 6-hourly and daily time series, to compute indices representative of intense and extreme conditions."

l21: "for average precipitation"

Sorry, I don't understand this point.

l26: "lost opportunities" What does this mean?

We removed this part of the sentence that in the new version is:

"An extreme climate event can have an impact on human activities, either as direct and indirect damages and, unfortunately also as loss of human life."

l29: "GCM simulations" -> Simulations of GCMs"

Done.

l178-179: This should be re-worded

Rewritten as:

"*This result suggests that a higher horizontal resolution is not sufficient to improve the representation of extreme temperature events at the highest time frequency considered. Consequently, the worsening of model biases in high frequency (6-hourly) temperature statistics derives from deficiencies of the current version of model components and parameterizations in representing high-frequency processes.*"

Figure 1: This may be an ignorant question, but I guess the 99th percentile could also be for negative temperature values. Whether you are looking into hot or cold temperatures should be specified somewhere.

Not sure to understand the point. This is the 99th percentile computed over temperature time series that can contain negative values too. With this said it can be that, at least at high latitudes, the 99th percentile is still negative, despite sitting on the right tail of the temperature distribution.

Figure 5: Please use [mm/d] and not [%].

Done.

---

## Author Comment (AC2)

**Answers to RC2:**
We thank the reviewer for the suggestions and we answered (in blue) to each single comment (in black), focusing on the best solution to satisfy RC2 and RC1 comments (both available on the online discussion).

Major comments
1. The introduction does not provide sufficient context to the study. In particular, the authors should expand it to include a discussion on prior studies which have looked at the impact of resolution on the ability of GCMs in simulating extreme events.
We thank the reviewer for this suggestion and in the new version of the manuscript we added the following part to the introduction, also supporting the discussion of the results presented in section 5:
*"Regarding the extreme temperature representation, based on data at the daily frequency, it has been shown that GCMs tend to have warm bias over most land areas (Li et al., 2021) and the horizontal resolution plays a minor role with respect to the one played in the extreme precipitation representation (Kharin et al. 2013; Wei et al., 2019). Typically the warm extremes are computed based on maximum daily temperature, but in this work we want to verify the potential improvements induced by the increased resolution in the representation of extreme temperature events defined at two different time frequency (daily and 6-hourly). For this reason we investigate the distribution of daily and 6-hour average temperature, instead of maximum daily temperature (Scoccimarro and Navarra, 2020).*

*Regarding the extreme precipitation representation, Based on simulations from single GCM, some improvement in skill at higher resolution for some measures of extreme precipitation over certain regions of the globe have been found in the past (Wehner et al. 2014, Kopparla et al. 2013) and only recently, multi-model assessment on this topic have been done, confirming that increasing the horizontal resolution to ¼ of degree (the highest adopted by the model object of this study), the magnitude of simulated daily (Bador et al. 2020) and sub-daily precipitation (Wehner et al. 2021) extremes is increased. On the other hand this is not associated to a systematic improvement in the simulation of precipitation extremes when compared to observations and, quantitatively, at the global scale, the intensification of precipitation extremes at increased resolution varies substantially from model to model (Bador et al. 2020). Also, for grid point GCMs (as opposed to spectral GCMs), the fraction of land precipitation increases, largely due to better resolved orography (Vannière et al., 2019; Terai et al., 2018; Demory et al., 2014)."*

Added References:
-Demory, M.-E., Vidale, P. L., Roberts, M. J., Berrisford, P., Strachan, J., Schiemann, R., and Mizielinski, M. S.: The role of horizontal resolution in simulating drivers of the global hydrological cycle, Clim. Dynam., 42, 2201–2225, https://doi.org/10.1007/s00382-013-1924-4, 2014.
-Kopparla, P., Fischer, E. M., Hannay, C., & Knutti, R.: Improved simulation of extreme precipitation in a high-resolution atmosphere model. Geophysical Research Letters, 40, 5803–5808. https://doi.org/10.1002/2013GL057866. 2013.

- Kharin, V.V., Zwiers, F.W., Zhang, X. *et al.* Changes in temperature and precipitation extremes in the CMIP5 ensemble. *Climatic Change* **119,** 345–357. https://doi.org/10.1007/s10584-013-0705-8. 2013.
- Li, C., Zwiers, F., Zhang, X., Li, G., Sun, Y., & Wehner, M.. Changes in Annual Extremes of Daily Temperature and Precipitation in CMIP6 Models, *Journal of Climate*, *34*(9), 3441-3460. 2021.
- Scoccimarro E., Navarra A.: Precipitation and temperature extremes in a changing climate. Chapter 2 in "Hydrometeorological Extreme Events and Public Health" Wiley book, 320 pages. ISBN: 978-1-119-25930-5. 2021
-Terai, C. R., Caldwell, P. M., Klein, S. A. Tang, Q., and Branstetter, M. L.: The atmospheric hydrologic cycle in the ACME v0.3 model, Clim. Dynam., 50, 3251–3279, https://doi.org/10.1007/s00382-017-3803-x, 2018.
-Vanniere, B., Vidale, P. L., Demory, M.-E., Schiemann, R., Roberts, M. J., Roberts, C. D., Matsueda, M., Terray, L., Koenigk, T., and Senan, R.: Multi-model evaluation of the sensitivity of the global energy budget and hydrological cycle to resolution, Clim. Dynam., 52, 6817–6846, https://doi.org/10.1007/s00382-018-4547-y, 2019
-Wehner MF et al.: The effect of horizontal resolution on simulation quality in the Community Atmospheric Model, CAM5.1. J. Adv. Model. Earth Syst. 6, 980–997. doi:10.1002/2013MS000276, 2014.
- Wei, L. X., X. G. Xin, C. Xiao, et al.: Performance of BCC-CSM Models with Different Horizontal Resolutions in Simulating Extreme Climate Events in China. *J. Meteor. Res.*, **33**(4): 720-733. doi: 10.1007/s13351-019-8159-1, 2019.

2. While I agree with using two products to evaluate the model's precipitation, I would have to disagree with the use of ERA5 for that purpose. Precipitation is not assimilated in reanalyses and is thus a product of the model used to create it. Although ERA is a superior product to its predecessor, there are many known issues with ERA5 precipitation. See for example:
Rivoire, P., Martius, O., & Naveau, P. (2021). A comparison of moderate and extreme ERA-5 daily precipitation with two observational data sets. Earth and Space Science, 8, e2020EA001633. https://doi.org/10.1029/2020EA001633
Hu, G., Franzke, C. L. E. (2020). Evaluation of daily precipitation extremes in reanalysis and gridded observation based data sets over Germany. Geophysical Research Letters, 47, e2020GL089624. https://doi.org/10.1029/2020GL089624
Crossett et al. (2020) Evaluation of Daily Precipitation from the ERA5 Global Reanalysis against GHCN Observations in the Northeastern United States. Climate, 8, 148; doi:10.3390/cli8120148
It would thus be better to use another observational product to evaluate the model.
In the new version of the manuscript we do not rely on ERA5 precipitation for model evaluation. Also following the Reviewer #1 comment on the same topic, we decided to use a new observational dataset (in addition to the already involved CHIRPS dataset) instead of ERA5. The MSWEP (Beck et al. 2019) dataset is a global precipitation product with a 3-hourly 0.1° resolution available at a 3-hourly temporal resolution, covering the period from 1979 to the near present. The dataset takes advantage of the complementary strengths of gauge-, satellite-, and reanalysis-based data to provide reliable precipitation estimates over the globe. With this dataset we compute seasonal averages and both daily and 6-hourly percentiles to

evaluate model results (same as done based on ERA5 in the previous version of the paper, but over a shorter period [1981-2014).

The three suggested references have been added to the text to justify the choice to do not use the ERA5 precipitation for comparison. This is the sentence added to the text:

"*Since there are many known issues with ERA5 precipitation (Rivoire et al., 2021; Hu et al., 2020; Crosset et al. 2020), for the evaluation of the model performance in representing the precipitation distribution we build on MSWEP version 2 observational data set (Beck et al. 2019): The Multi-Source Weighted-Ensemble Precipitation (MSWEP) global precipitation dataset is available at a 3-hourly temporal resolution, covering the period from 1979 to the near present, with an horizontal resolution of 0.1 degrees. The dataset takes advantage of the complementary strengths of gauge-, satellite-, and reanalysis-based data to provide reliable precipitation estimates over the globe.*"

Added References:
-Beck, H. E., Wood, E. F., Pan, M., Fisher, C. K., Miralles, D. G., van Dijk, A. I. J. M., McVicar, T. R., & Adler, R. F.. MSWEP V2 Global 3-Hourly 0.1° Precipitation: Methodology and Quantitative Assessment, Bulletin of the American Meteorological Society, 100(3), 473-500, 2019.
-Rivoire, P., Martius, O., & Naveau, P. (2021). A comparison of moderate and extreme ERA-5 daily precipitation with two observational data sets. Earth and Space Science, 8, e2020EA001633. https://doi.org/10.1029/2020EA001633
-Hu, G., Franzke, C. L. E. (2020). Evaluation of daily precipitation extremes in reanalysis and gridded observation based data sets over Germany. Geophysical Research Letters, 47, e2020GL089624. https://doi.org/10.1029/2020GL089624
-Crossett et al. (2020) Evaluation of Daily Precipitation from the ERA5 Global Reanalysis against GHCN Observations in the Northeastern United States. Climate, 8, 148; doi:10.3390/cli8120148

3. I am not sure that daily mean 2m temperature is the best variable to evaluate extreme temperatures in the model. Why not use tasmin and tasmax? And Figure 3 could be sent to the supplementary material as it is hardly discussed.

We agree that looking at the distribution of daily tasmax is different from looking at the distribution (and relative tails) of average daily temperature, but the approach used in the current manuscript (also used in Scoccimarro and Navarra, 2021) has some advantages such as the fact that tasmax parameter depends on the model time step length (different in the different versions of the model) while average temperature (daily or 6-hourly) is independent from the model time step. Also, the usage of values averaged over a period (daily or 6-hourly) instead of tasmax gives an information more exhaustive from the human health perspective: e.g. a few minutes (model time step) with 42°C might be less problematic for the human body than 6 hours at 38 °C.

In addition, since tasmax is defined only at the daily frequency (this is true for all of the CMIP5 and CMIP6 model output available on ESGF), it is impossible to compare the model horizontal resolution role in representing daily and 6-hourly statistics.

Last but not least, we recently retrieved CMCC-HR4 and CMCC-VHR4 tasmax and tasmin fields from the ESGF repository because we found a bug on both these daily datasets.

4. The color schemes used to present the data makes it difficult to understand the results. For one, it saturates very quickly. For example, on Fig. 1, it is nearly impossible to distinguish values between -6 and -20 (when it is printed on paper). And also, there are similar colors on both sides of the 0 point (e.g. green on Fig 3.). It made reading through the precipitation subsection particularly difficult, as I couldn't get a good sense of the size of the biases that were being shown. My suggestion would be to refer to the IPCC visual style guide:

https://www.ipcc.ch/site/assets/uploads/2019/04/IPCC-visual-style-guide.pdf

Following this suggestion, In the new version of the manuscript color schemes have been defined following the IPCC visual style guide. We also had to modify the structure of the figures to put the model fields, the reference and the bias fields into the same figure, dividing 6h/24h and DJF/JJA to follow reviewer #1 request.

5. I think the manuscript would benefit from an attempt at explaining some of the results that are presented. The authors described the convection scheme in Section 2.1, because "it is worthwhile to mention for our discussion on precipitation biases", but the convection scheme is never referred to when the results are presented. Does it explain the differences between results obtained with 6-hourly data and daily mean? And if so, how? Could dry biases in the model play a role in the extreme of near surface temperature? Was the impact of resolution on extreme temperature and precipitation evaluated by other groups using the CAM model? Are the results consistent with those found here? Furtheremore, Vaniere et al. (2019) has shown a significant impact of resolution on precipitation over mountainous areas in HighResMIP models. Are the results presented here consistent with that study and others that have looked into this issue previously? Given that this is a single model study, it is difficult to evaluate if the results are model dependent. Expanding the discussion would help in that regard.

Vaniere et al. (2019) Multi-model evaluation of the sensitivity of the global energy budget and hydrological cycle to resolution. Climate Dynamics, 52, 6817–6846

We extended the description of the convection scheme in the standard version of CAM4 comparing it to the one adopted by CAM5:

"*In other words the deep convection scheme is triggered based on a minimum positive threshold of CAPE, same as in the standard resolution of the CAM5 model (Wang and Zhang, 2013).*"

And this is supporting the added text in the discussion:

"*The high-resolution version of the model generates excess extreme precipitation in the wet, warm regions, or seasons, consistently with findings based on experiments carried out with the CAM5 atmospheric model at the same resolutions (Wehner et al, 2014), highlighting once again the importance of an extensive model tuning at the high resolution*".

The differences between extreme precipitation biases in 6h and daily data moving from standard to high resolution is not that evident, thus we didn't link this to the description of the precipitation parameterization.

Regarding the role of dry biases we assume that this comment is related the 99p bias since the average precipitation (S3 and S4 figures in the new version of the manuscript) tends to show a wet bias. A first investigation on the role of such dry bias in modulating extreme near surface temperature, does not suggest a systematic relationship: the bias of the 99p of precipitation during summer (Figure 6) is dry for the low resolution model but wet for the

high resolution model over part of the Maritime Continent and South America, while the bias in 99p of near surface temperature is positive for both models, over the same regions (Figure 4). During winter, for both models, the most pronounced positive biases in 99p of temperature (Figure 3) are over regions where the bias in the 99p of precipitation is negligible (Northern Hemisphere north of 70ºN) and over South America where the bias in the 99p of precipitation is positive in the standard resolution version and negative in the high resolution version (Figure 5).

The work by Vanniere et al. (2019) has been now mentioned in the introduction (see the answer to your first comment) and a comment on the CMCC-CM2 model results within the Venniere et al. analysis is provided in the conclusion as:

*"In addition it is important to note that moving from the standard to the high resolution of CMCC-CM2, the model behaves as most of the models participating to the HighResMIP project with the tendency to an increased fraction of land precipitation in the highest resolution, same as for the fraction of land precipitation caused by moisture convergence increased with resolution (Venniere et al. 2019). Also, in CMCC-CM2 model, the orographic precipitation captures most of the change of precipitation due to resolution, consistently with most of HighResMIP models (Venniere et al. 2019)"*.

Added references:
- -Vanniere, B., Vidale, P. L., Demory, M.-E., Schiemann, R., Roberts, M. J., Roberts, C. D., Matsueda, M., Terray, L., Koenigk, T., and Senan, R.: Multi-model evaluation of the sensitivity of the global energy budget and hydrological cycle to resolution, Clim. Dynam., 52, 6817–6846, https://doi.org/10.1007/s00382-018-4547-y, 2019
- Wang X., Zhang M.: An analysis of parameterization interactions and sensitivity of single-column model simulations to convection schemes in CAM4 and CAM5. Journal of Geoph. Research Atm. https://doi.org/10.1002/jgrd.50690. 2013.
- Wehner MF et al.: The effect of horizontal resolution on simulation quality in the Community Atmospheric Model, CAM5.1. J. Adv. Model. Earth Syst. 6, 980–997. doi:10.1002/2013MS000276, 2014.

Minor comments
CMCC-CM2-HR and CMCC-CM2-VHR might be the name of the models, but it is strange to refer to a model with a resolution of 1 deg (for the atmospheric component) as high resolution. It might be easier for the reader to simply refer to the two configurations as standard resolution (1 deg) and high resolution (0.25 deg). To be clear, I am not suggesting changing the name of the models, but simply to use the terms standard resolution and high resolution (or something along those lines) when referring to MCCCM2-HR and MCC-CM2-VHR.
Done: In the new version of the manuscript we use the terms "standard" and "high" instead of "high" and "very-high".

The authors should mention the name of the experiment from which the data are taken. Vaniere et al. (2019) noted different responses in terms of the impact of resolution on precipitation between grid point and spectral models. As such, the type of atmospheric model should be highlighted and the authors should mention whether their results are consistent with that prior study.

We added a sentence in 2.1 section to indicate the experiment from which the data are taken: *"In the current analysis we investigate the hist-1950 HighResMIP experiment as described in section 2.3.".*
Also we made explicit the grid point configuration in section 2.1:
*"The CMCC general circulation has been developed in several configurations (Cherchi et al. 2019). The model uses as atmospheric component the CAM Atmospheric component (CAM4, Neale et al. 2010) in its grid point configuration"*
A comparison of CMCC-CM2 model results to other HigResMIP results, based on Venniere et al. analysis, is now part of the discussion (see the answer to your last major comment).

p.1, line 26: "A climate variation can have an impact on human activities...". I am not sure what the authors mean by "climate variations: in this context, but this phrasing is a bit odd. I would suggest rewriting.
Rewritten as:
*"An extreme climate event can have an impact on human activities, either as direct and indirect damages and, unfortunately also as loss of human life."*

p. 1, line 27: "Extreme climate events are involved in the vast majority of the most severe episodes." The most severe episodes of what?
The sentence has been removed also in accord with the rephrasing of the previous one.

p. 2, line 32: "was designed to understand the role of the horizontal resolution."
The role of horizontal resolution on what?
Rewritten as:
*" .. was designed to understand the role of the horizontal resolution in improved process representation in all components of the climate system"*

p.2, line 33: "based on two versions of the GCM"
Done.

p. 2, line 34: " differing only in their atmospheric horizontal resolution"
Done.

p.2 line 41: "However, such analyses has employed rather low frequency data…"
I am not sure what analyses the authors are referring to (or what they mean by low and high frequency), but many studies have used daily or sub-daily data to look at extremes in climate models. See for example:
Wehner M, Lee J, Risser M, Ullrich P, Gleckler P, Collins WD. 2021 Evaluation of extreme sub-daily precipitation in high-resolution global climate model simulations.
Phil.Trans.R.Soc.A379: 20190545. https://doi.org/10.1098/rsta.2019.0545
Wehner MF et al. 2014. The effect of horizontal resolution on simulation quality in the Community Atmospheric Model, CAM5.1. J. Adv. Model. Earth Syst. 6, 980–997. doi:10.1002/2013MS000276.
And references therein.
The sentence has been modified as;

*"However, most of the analyses on extreme events employ rather low frequency data (tipically daily), and short-duration high-intensity precipitation events can easily escape detection if high-frequency data are not used (Meredith et al. 2020, Scoccimarro et al. 2015)."*
The two mentioned references are part of the manuscript and helped to improve the manuscript based on the answer to the major comments of the reviewer, see for instance new line 70:
*"..and only recently, multi-model assessment on this topic have been done, confirming that increasing the horizontal resolution to ¼ of degree (the highest adopted by the model object of this study), the magnitude of simulated daily (Bador et al. 2020) and sub-daily precipitation (Wehner et al. 2021) extremes is increased "*
and new line 507:
*"In principle, horizontal resolution increases should improve the representation of extreme storms such as tropical cyclones (Scoccimarro et al. 2020) and for this reason also the representation of the associated short term extreme precipitation should improve, but this is not the case for the model object of this study, and it is also confirmed by recent analysis on the same topic (Wehner et al., 2021)."*

p.2 line 62: "The two models object of this study..   degree in VHR."
I would recommend moving this sentence to the previous paragraph when the authors discuss the atmospheric component of the model.
Done.

p.4, line 115: "Also, the positive HR DJF bias over eastern Europe is more than halved in VHR". To me, it seems like it disappears, but it might be due to the colorbar.
This is now more clear based on the new color scheme proposed (see new Figure 1 lower panels).

p.4, line 118: "The positive extreme temperature bias between 30N and 60N shown by the HR model during JJA is partially reduced in VHR." This seems to happen mostly over Europe and Asia, not so much North America.
Rewritten as:
*"The positive extreme temperature bias between 30°N and 60°N shown by the HR model during JJA (Figure 2 lower left panel) is partially reduced in VHR especially over Europe and Asia."*

p.4, line 119: "the 5 to 7C positive JJA bias over the western coast of South America in HR results haved in HR". That might be the case, but it is really hard to see in the figure.
Also, some words seem to be missing in that sentence.
I'm not sure about the source of the aforementioned phrase, because it seems different from what I see in the submitted manuscript, but anyway, the sentence in the new version of the manuscript is:
"Similarly, the 5 to 7°C positive JJA bias over the western coast of South America in HR, results halved in VHR"
We think that the new color scheme is more appropriate (see new figure 2)

p.5, line 129: "the model extreme precipitation is compared to…"

Done.

p.5, lines 155-164. I have to confess I didn't quite understand that explanation.
This part has been partially rephrased as follow:
"*The worsening of the extreme precipitation bias moving from the HR to the VHR model along the tropics, especially in the Southern Hemisphere during JJA, is also associated to a deterioration of the representation of the fraction of precipitation associated to extreme events with respect to the total precipitation: this metric is obtained accumulating the water of all the events more intense than the 99p, and normalizing it by the total amount of precipitation in the considered period (season by season). Figure S17 shows that both models reasonably well capture this metric in both seasons compared to MSWEP, but the VHR model tends to overestimate such amount over the southern Hemisphere, except for the Australian domain. In particular, the strong positive bias of DJF average precipitation over Australia (up to 4 mm/d, Figure S3, lower panels) can't be attributed to the positive (higher than 15 mm/d, Figure 5 lower panels) bias found for extreme events, but must be associated to a right shift of the remaining part of the precipitation distribution, more pronounced for the non-extreme events as partially confirmed by the positive bias in the 90p metric over the same season (Figure S11)*"

p.6, line 188: Replace PRIMAVERA by HighResMIP
Done.

---

## Referee Report (RR1)

This is the second time that I am reviewing this manuscript. The manuscript is improved with respect to the first version and most of my previous comments were successfully addressed (two were not entirely, see items 1 and 2 below). I also added a few new (most relatively minor) comments derived from the new material that was added (items 3-6). Finally, I would recommend the manuscript be looked over by a native English speaker, as I found many English mistakes in the document. I have noted a few at the end, but this is not an exhaustive list, and the last section was a bit difficult to understand (see item 6).

1. Figures: The authors are using a diverging palette for non diverging data (upper right in Figures 1-10. While I don't think it is such a problem for temperature in this case, for precipitation it gives the illusions of boundaries where there is none. I would suggest using a sequential scheme, as described on p.10-11 of the IPCC visual style guide.

Also, there are a lot of figures for such a short manuscript. Since the emphasis is on the biases, may I suggest the following format to reduce their number?

Merge Fig. 1-4 such that:

| ERA5 DJF tas 24h | HR DJF bias 24h | VHR DJF bias 24h |
| ERA5 DJF tas 6h | HR DJF bias 6h | VHR DJF bias 6h |
| ERA5 JJA tas 24h | HR JJA bias 24h | VHR JJA bias 24h |
| ERA5 JJA tas 6h | HR JJA bias 6h | VHR JJA bias 6h |

and send the absolute values for the simulations (i.e. the second rows of the current figures) to the supplementary section. Figures 5-8 could be merged in a similar way. And Figures 9-10 could probably be sent to the supplementary section given that they are so similar to Figures 7-8 and only 1 sentence is spent discussing them. If the authors would like to include a 3rd picture, they could include what is currently Figure S17.

Finally, the characters for the latitudes and longitudes on certain figures seem to be overlapping (see for example Figure 5).

2. Something that was pointed out by myself and the other reviewer was the use of high and very high resolution when referring to the two configurations. In their response to the reviewers, the authors mention "In the new version of the manuscript we use the terms standard and high instead of high and very-high". However, I found many instances where the authors refer to the 1deg resolution simulation as high resolution (HR) and the 0.25deg resolution as very high-resolution (VHR). e.g. lines 34-35, 41, 84, 90, etc.

3. Lines 52-56:" Regarding the extreme precipitation representation, based on simulations from single GCM, some improvement in skill at higher resolution for some measures of extreme precipitation over certain regions of the globe have been found in the past (Wehner et al. 2014,

Kopparla et al. 2013) and only recently, multi-model assessment on this topic have been done, confirming that increasing the horizontal resolution to ¼ of degree (the highest adopted by the model object of this study), the magnitude of simulated daily (Bador et al. 2020) and sub-daily precipitation (Wehner et al. 2021) extremes is increased."

I feel this ignores all the work done on this topic in the RCM community, and it somewhat contradicts what is written on line 40: ("high resolution models, when implemented with a resolution similar to VHR, achieve skills comparable to state-of-the-art Regional Climate Models in reproducing precipitation distributions"). I would suggest reformulating.

4. Line 151: "On the other hand, the negative JJA bias of about -8C over north-eastern Canada shown by **the** HR model is even worse in the VHR version, where a larger portion of the domain is subject to a bias of about 10C."

This negative bias sticks out like a sore thumb. Do the authors have an explanation for this very large negative bias? Is it linked to an excess of sea ice in the summer?

5. I recommend computing the mean bias for the various variables and the different configurations. That value could be inserted directly in the corner of the relevant figure. That would help supporting statements such as:

"In terms of average precipitation, the VHR model shows less pronounced biases with respect to **the** HR model…(line 171)

"on average, the highest resolution CMCC model is better than the lower resolution model in representing…" (line 213)

6. I have to confess that I had a difficult time understanding the conclusion section, from line 225 onward. Part of it might be due to the English I think (e.g. sentence on lines 238-242 is too long; the following sentence is missing a verb), but some of the sentences in that section that were added as answers to the reviewers comments  are not really well integrated in the text, which makes the message a little confusing. I am afraid I don't have a good suggestion here, other than spending a bit of time to make sure that the text flows a little better and making sure the main conclusions/ideas are clearly put forth.

**Minor points**

Line 40: "Demory et al. (2020) have shown that high-resolution models"

Line 43: typically, not tipically

Line 45-47: "Regarding the extreme temperature representation, based on data at the daily frequency, it has been shown that GCMs tend to have warm bias over most land areas (Li et al.,

2021) and the horizontal resolution plays a minor role with respect to the one played in the extreme precipitation representation"

1) I would suggest rephrasing this sentence, as the wording is a bit awkward.
2) Do you mean to say that models overestimate both warm and cold extremes?
3) Play a minor role in what?

Lines 47-51: "Typically, the warm extremes are computed based on maximum daily temperature, but in this work we want to verify the potential improvements induced by the increased resolution in the representation of extreme temperature events defined at two different time frequency (daily and 6-hourly). For this reason we investigate the distribution of daily and 6-houry average temperature, instead of maximum daily temperature."

This should be moved to the methodology section.

Line 53: "...based on simulations from a single GCM…"

Line 82: "The two models object of this study differ only…"

Line 95: "The model performance in representing the temperature distribution is evaluated by comparing results to…"

Line 109: "Since we aim to characterize different types of extreme events…"

Line 119:" This time period is sufficiently long to capture…"

Line 122: "The grid differences are minor and therefore the interpolation introduces very little differences in the fields."

I don't know if I agree with this statement. It is true that the difference is small between the VHR resolution and ERA 5, but CHIRPS resolution is 0.05deg while the atmospheric models used here have a 0.25 deg and 1 deg resolution.

Line 155: "The positive bias over the north western part of South America…"

Line 184: "suggesting that the worst VHR extreme precipitation representing during DJF is mainly due to a too much pronounced stretching of the right part of the precipitation distribution only"

Please rephrase.

Line 213: "Anyway, on average…"

Line 227: "...and also with multi-model recent findings, suggesting that higher resolution models…"

I don't quite understand what the authors mean with "and also with multi-model recent findings".

---

## Author Response (AR2)

We thank the reviewers for their additional suggestions for minor revision and we answered (in blue) to each single comment (in black).

**Answers to RC1:**

Many thanks for the detailed revisions of the paper. The paper is ready to be accepted after very minor revisions.

We are happy to hear that the present version of the manuscript has been appreciated.

l. 27: Remove "then"

Done.

l. 43: tipically

Done.

You have stopped using "high resolution" and "very high resolution" but continued to use "HR" and "VHR". This is confusing and should be changed.

This is to maintain the reference to the model version name. This choice is in accord with a comment received in the previous round of review.

l. 116: It seems a bit strange to start a new section between the previous and the following paragraph. Maybe change the position of the section heading?

The present organization of subsections puts together model data (2.1) observational data sets (2.2) and the methodological details used for our analysis (2.3). Moving the first paragraph of 2.3 to 2.2 would result in a mix between data and method description. We would prefer to maintain the current structure.

l.181: "compared ot"

Done.

l.206: "for the models"

Done.

ll.235: Does this maybe indicate that you get the daily cycle wrong? -> Again, changes in the parametrisation may help?

Yes, it might also be related to the fact that the daily cycle is not well capture over certain regions, but we can't claim it in the current version: additional analysis would be necessary.

l.242: "worse"

Done.

l.246-247: Should be reworded.

Rewritten as: "*In addition it is important to note that moving from the standard to the high resolution of CMCC-CM2, the model behaves consistently with the models participating to the HighResMIP project: a tendency to an increased fraction of land precipitation in the highest resolution, and the same tendency for the fraction of land precipitation caused by moisture convergence (Venniere et al. 2019).*"

This is the second time that I am reviewing this manuscript. The manuscript is improved with respect to the first version and most of my previous comments were successfully addressed (two were not entirely, see items 1 and 2 below). I also added a few new (most relatively minor) comments derived from the new material that was added (items 3-6). Finally, I would recommend the manuscript be looked over by a native English speaker, as I found many English mistakes in the document. I have noted a few at the end, but this is not an exhaustive list, and the last section was a bit difficult to understand (see item 6).

We thank the reviewer for this second round of comments. We answered to each specific comment (see below), also improving the language.

1. Figures: The authors are using a diverging palette for non diverging data (upper right in Figures 1-10. While I don't think it is such a problem for temperature in this case, for precipitation it gives the illusions of boundaries where there is none. I would suggest using a sequential scheme, as described on p.10-11 of the IPCC visual style guide.

Figures related to precipitation have been modified following this suggestion.

Also, there are a lot of figures for such a short manuscript. Since the emphasis is on the biases, may I suggest the following format to reduce their number?
Merge Fig. 1-4 such that:
ERA5 DJF tas 24h HR DJF bias 24h VHR DJF bias 24h
ERA5 DJF tas 6h HR DJF bias 6h VHR DJF bias 6h
ERA5 JJA tas 24h HR JJA bias 24h VHR JJA bias 24h
ERA5 JJA tas 6h HR JJA bias 6h VHR JJA bias 6h
and send the absolute values for the simulations (i.e. the second rows of the current figures) to
the supplementary section. Figures 5-8 could be merged in a similar way. And Figures 9-10 could probably be sent to the supplementary section given that they are so similar to Figures 7-8 and only 1 sentence is spent discussing them. If the authors would like to include a 3rd picture, they could include what is currently Figure S17.

The number of figures was lower in the first version of the manuscript but we had to modify it (leading to the current structure) to match reviewer 1 requests.

Finally, the characters for the latitudes and longitudes on certain figures seem to be overlapping
(see for example Figure 5).

Corrected.

2. Something that was pointed out by myself and the other reviewer was the use of high and very high resolution when referring to the two configurations. In their response to the reviewers, the authors mention "In the new version of the manuscript we use the terms standard and high instead of high and very-high". However, I found many instances where the authors refer to the 1deg resolution simulation as high resolution (HR) and the 0.25deg resolution as very high-resolution (VHR). e.g. lines 34-35, 41, 84, 90, etc.

We adopted "standard" and "high" as suggested, but we preferred to maintain the original acronyms (HR and VHR) for consistency with the model name in the CMIP6 repository, since

there is also a SR version provided by CMCC, but not used in the present study. This choice is in accord with a comment by reviewer 1 received in the previous round of reviews.

3. Lines 52-56:" Regarding the extreme precipitation representation, based on simulations from single GCM, some improvement in skill at higher resolution for some measures of extreme precipitation over certain regions of the globe have been found in the past (Wehner et al. 2014, Kopparla et al. 2013) and only recently, multi-model assessment on this topic have been done, confirming that increasing the horizontal resolution to ¼ of degree (the highest adopted by the model object of this study), the magnitude of simulated daily (Bador et al. 2020) and sub-daily precipitation (Wehner et al. 2021) extremes is increased."
I feel this ignores all the work done on this topic in the RCM community, and it somewhat contradicts what is written on line 40: ("high resolution models, when implemented with a resolution similar to VHR, achieve skills comparable to state-of-the-art Regional Climate Models in reproducing precipitation distributions"). I would suggest reformulating.
We now specify in the second sentence that this statement (line 52-56) is related to GCMs: "Regarding the extreme precipitation representation in GCMs"

4. Line 151: "On the other hand, the negative JJA bias of about -8C over north-eastern Canada shown by the HR model is even worse in the VHR version, where a larger portion of the domain is subject to a bias of about 10C."
This negative bias sticks out like a sore thumb. Do the authors have an explanation for this very large negative bias? Is it linked to an excess of sea ice in the summer?
We don't have a definitive explanation for this bias, but we confirm that the two CMCC-CM2 model versions object of this study tend to overestimate the sea ice over the Northern Hemisphere during JJA. We added a comment on this in the new version:
"This negative bias is also consistent with the tendency of the two versions of the CMCC-CM2 model to overestimate the sea ice cover during summer over the Northern Hemisphere (not shown)"

5. I recommend computing the mean bias for the various variables and the different configurations. That value could be inserted directly in the corner of the relevant figure. That would help supporting statements such as:
"In terms of average precipitation, the VHR model shows less pronounced biases with respect to the HR model…(line 171) "on average, the highest resolution CMCC model is better than the lower resolution model in representing…" (line 213)
Done: In the new version of the manuscript, within the figure panels referring to the biases, there is also the indication of the average value.

6. I have to confess that I had a difficult time understanding the conclusion section, from line 225 onward. Part of it might be due to the English I think (e.g. sentence on lines 238-242 is too long; the following sentence is missing a verb), but some of the sentences in that section that were added as answers to the reviewers comments are not really well integrated in the text, which makes the message a little confusing. I am afraid I don't have a good suggestion here, other than spending a bit of time to make sure that the text flows a little better and making sure the main conclusions/ideas are clearly put forth.
Long sentences have been divided and the section has been then improved as follow (see also the provided version of the manuscript with "track-change on"):

"Regarding the precipitation distribution, the VHR model performs better in representing averages and intense events, but more pronounced biases appear in VHR compared to HR when focusing on extreme events, with a more evident degradation in the daily statistics compared to the 6-hourly. This latter result reduces the confidence we usually attribute to the highest horizontal resolution in modelling extreme precipitation, and is consistent with single model analysis based on CAM5.1 atmospheric model (Wehner et al. 2014) suggesting a positive bias over most of the globe in the representation of extreme events at ¼ degree horizontal resolution. This is also in agreement with recent findings (Bador et al. 2020) suggesting that highest resolution models tend to produce more pronounced extremes than lower resolution ones. In addition many of them show lower skill in representing observed patterns, both in terms of intensity and spatial distribution, at the higher resolution, compared to their corresponding lower resolution version.

This emphasizes the need to focus not only on the horizontal resolution to improve the model ability in representing the climate system, but also on physics and tuning. It is important to note that in the model object of this analysis the tuning parameters were kept constant, moving from the HR to the VHR version, in order to be compliant with the HigResMIP protocol. The different biases, obtained based on daily and 6-hourly time frequencies, also suggest that for the setup of model physics and tuning we need to consider the event distributions at different time frequencies, to take into account the representation of the different processes responsible of the extreme conditions emerging at the different frequencies (Scoccimarro et al. 2015).

The poor performance of climate models in representing extreme precipitation is not improved in the last CMIP6 generation models, compared to the previous CMIP5 generation (Scoccimarro et al. 2020). In the present work we have shown that this lack is even more evident moving to the highest resolution version of the CMCC-CM2 model adopted for HighResMIP, consistently with multi-model analysis performed at the same horizontal resolution (Bador et al. 2020): GCMs whose parameterizations were not retuned at higher resolution lead to worse results. The high-resolution version of the model tends to overestimate extreme precipitation in the wet and warm regions, consistently with findings based on experiments carried out with the CAM5 atmospheric model at the same resolutions (Wehner et al, 2014), highlighting once again the importance of an extensive model tuning at the high resolution. In addition it is important to note that moving from the standard to the high resolution of CMCC-CM2, the model behaves consistently with the models participating to the HighResMIP project: a tendency to an increased fraction of land precipitation in the highest resolution, and the same tendency for the fraction of land precipitation caused by moisture convergence (Venniere et al. 2019). Also, in CMCC-CM2 model, the orographic precipitation captures most of the change of precipitation due to resolution, consistently with most of HighResMIP models (Venniere et al. 2019)."

Minor points
Line 40: "Demory et al. (2020) have shown that high-resolution models"
Done.
Line 43: typically, not tipically
Done.
Line 45-47: "Regarding the extreme temperature representation, based on data at the daily

frequency, it has been shown that GCMs tend to have warm bias over most land areas (Li et al.,2021) and the horizontal resolution plays a minor role with respect to the one played in the

extreme precipitation representation"

1) I would suggest rephrasing this sentence, as the wording is a bit awkward.

We rephrase the sentence as follow:

"Regarding the extremely high temperature representation, based on data at the daily frequency, it has been shown that GCMs tend to have warm bias over most land areas (Li et al., 2021) and the horizontal resolution plays a minor role in affecting the bias, with respect to the one played in the extreme precipitation representation (Kharin et al. 2013, Wei et al. 2019)."

2) Do you mean to say that models overestimate both warm and cold extremes?

I meant warm extremes, now specified.

3) Play a minor role in what?

The horizontal resolution plays a minor role in affecting the bias, as now specified.

Lines 47-51: "Typically, the warm extremes are computed based on maximum daily temperature, but in this work we want to verify the potential improvements induced by the increased resolution in the representation of extreme temperature events defined at two different time frequency (daily and 6-hourly). For this reason we investigate the distribution of daily and 6-houry average temperature, instead of maximum daily temperature."
This should be moved to the methodology section.

Done.

Line 53: "...based on simulations from a single GCM…"

Done.

Line 82: "The two models object of this study differ only…"

Sorry, I don't see the typo. The sentence is as you suggest in the manuscript too.

Line 95: "The model performance in representing the temperature distribution is evaluated by comparing results to…"

Done.

Line 109: "Since we aim to characterize different types of extreme events…"

Done.

Line 119:" This time period is sufficiently long to capture…"

Done.

Line 122: "The grid differences are minor and therefore the interpolation introduces very little differences in the fields."

I don't know if I agree with this statement. It is true that the difference is small between the VHR resolution and ERA 5, but CHIRPS resolution is 0.05deg while the atmospheric models used here have a 0.25 deg and 1 deg resolution.

We agree. This sentence was acceptable in the first version of the manuscript where we didn't use MSWEP 0.1degree data. Anyway, the interpolation is done only to compute biases on the

final fields, and not before the percentile analysis as stated: comparable results are obtained using bicubic interpolation. This is now added to the text as:

"The kind of  interpolation introduces very little differences in the fields (not shown)."

Line 155: "The positive bias over the north western part of South America…"
Done.

Line 184: "suggesting that the worst VHR extreme precipitation representing during DJF is mainly due to a too much pronounced stretching of the right part of the precipitation distribution only"
Please rephrase.
Rewritten as:
"… suggesting that the bad representation of  DJF extreme precipitation in VHR (Figure 5) is mainly due to a too much pronounced stretching of the right part of the precipitation distribution."

Line 213: "Anyway, on average…"
Done.

Line 227: "...and also with multi-model recent findings, suggesting that higher resolution models…"
I don't quite understand what the authors mean with "and also with multi-model recent findings".
It was just to highlight the multi-model nature of the cited work. Anyway, we use "recent findings" instead of  "multi-model findings" in the new version of the manuscript.